

# The AquiFR hydrometeorological modelling platform as a tool for improving groundwater resource monitoring over France: evaluation over a 60 year period.

Jean-Pierre Vergnes[1], Nicolas Roux[2], Florence Habets[3,4], Philippe Ackerer[5], Nadia Amraoui[1], François Besson[6], Yvan Caballero[1], Quentin Courtois[7], Jean-Raynald de Dreuzy[7], Pierre Etchevers[6], Nicolas Gallois[8], Delphine J. Leroux[2], Laurent Longuevergnes[7], Patrick Le Moigne[2], Thierry Morel[9], Simon Munier[2], Fabienne Regimbeau[6], Dominique Thiéry[1], Pascal Viennot[8]

[1]BRGM French Geological Survey, Water Environment and Ecotechonologies Division, Orléans Cedex 2, France
[2]GAME/CNRM, Météo France, CNRS, Toulouse, France
[3]CNRS/Sorbonne University, UMR 7619 Métis, Paris Cedex 5, France
[4]Laboratoire de Géologie, Ecole normale supérieure, PSL Research University, CNRS UMR 8538, 24 rue Lhomond, 75005 Paris, France
[5]LHYGES, UMR 7517 CNRS, EOST / Université de Strasbourg, France
[6]METEO-FRANCE/DCSC, Toulouse, France
[7]Géosciences Rennes, UMR 6118, CNRS, Université de Rennes I, Rennes, France
[8]Centre de Géosciences, MINES ParisTech, Fontainebleau, France
[9]CERFACS, Toulouse, France

*Correspondence to*: Jean-Pierre Vergnes (jp.vergnes@brgm.fr)

**Abstract.** The new AquiFR hydrometeorological modelling platform was developed to provide short to long-term forecasts for groundwater resource management in France. The present study aims to describe and assess this new tool over a long-term period of 60 years. This platform gathers in a single numerical tool different hydrogeological models covering much of the French metropolitan area. Eleven aquifer systems are simulated through spatially distributed models using either the MARTHE groundwater modelling software or the EauDyssée hydrogeological platform. Twenty-three karstic systems are simulated by lumped models using the EROS software. AquiFR computes the groundwater level, the groundwater surface water exchanges, and the river flows at multiple river gauging stations. A simulation covering a 60 year period from 1958 to 2018 is achieved in order to evaluate the performance of this platform. The 8 km resolution SAFRAN meteorological reanalysis provides the atmospheric variables needed by the SURFEX land surface model in order to compute surface runoff that are used by all the hydrogeological models. The assessment is based on a wide range of selected piezometers as well as gauging stations corresponding to simulated rivers and outlets of karstic systems. For the simulated piezometric heads, 40% and 60% of the absolute biases are lower than 2 m and 4 m respectively. The Standardized Piezometric Level Index (SPLI) was computed to assess the ability of AquiFR to identify extreme events such as groundwater flooding or droughts in long-term simulations over a set of piezometers used for groundwater resource management. 55% of the Nash-Sutcliff scores calculated between the observed and simulated SPLI time series are greater than 0.5. Further work will focus on the use of this platform for short-term to seasonal forecasts in an operational mode and for climate change impact assessment.



# 1 Introduction

Groundwater is the most important freshwater resources on the Earth, but it is still poorly known. Groundwater is indeed located at some depth below the soil and therefore this resource is mostly monitored through well networks that can give information only at specific locations (Aeschbach-Hertig and Gleeson, 2012; Fan et al., 2013). Remote sensing gravimetry can provide large scale estimates of groundwater storage changes (Long et al., 2015) but it is not suited for regional scale studies (Longuevergne et al., 2010). Thus, modeling is still a useful tool to provide meaningful information on the groundwater resources (Aeschbach-Hertig and Gleeson, 2012).

An increasing number of numerical weather prediction models includes a representation of groundwater (Barlage et al., 2015; Sulis et al., 2018). Nevertheless, such representations are not detailed enough to be used to monitor or to forecast the groundwater resources. This is the reason why some dedicated approaches aim at providing groundwater level forecasts at the well scale with lumped models (Prudhomme et al., 2017) or neural networks (Amaranto et al., 2018; Dudley et al., 2017; Guzman et al., 2017).

At the regional scale, only a few modelling approaches use spatially distributed models to monitor and forecast the groundwater resource. Henriksen et al. (2003) presented the development of national hydrogeological models in Denmark aiming at gathering competencies from research organisms and water agencies and establishing a national overview of the present state and future trends of groundwater resources. An integrated groundwater/surface water hydrological model covering a spatial extension of about 43 000 km$^2$ with a 1 km grid resolution was then developed by making full use of the available data (Henriksen et al., 2003). The modelling system is composed of 11 regional sub-models. This model has been regularly updated, as reported by Højberg et al. (2013) who used local studies in relation with active stakeholders to include local data to improve the national model. The Danish model is planned to be used for real time monitoring (He et al., 2016) and climate change studies (Højberg et al., 2013).

In the Netherlands, national and regional water authorities decided to build the Netherlands Hydrological Instrument (NHI) which couples various physical models for all parts of the water system in order to support long-term plans for sustainable water use and safety under changing climate conditions (De Lange et al., 2014). The model was developed by research institutions but local knowledge has been adopted in cooperation with the national water boards (Højberg et al., 2013). It aims to be a model for long-term national policymaking and real-time forecasting for daily water management.

In the United Kingdoms (UK), Pachocka et al. (2015) used a numerical model representing the three most important UK aquifers as separate layers discretized using a 5 km resolution grid with a finite-difference scheme. The layers are connected to the river network and the model was tested against 37 gauging stations distributed across the country. Good fit to the observations was obtained in a steady state run. This study seems to be the first step toward a system that will be used for water management studies and climate impact studies.

Another study covering a wide domain corresponding to the major part of the United States (6.3 billion of km$^2$) was carried out by Maxwell et al. (2015). A 3-dimensional hydrogeological model (ParFlow) was used at a 1 km grid resolution in



steady state run. This model has four layers over the first meter of soil, and then a fifth layer from 1 to 100 meter depths. The computation time was one week on high-performance computer for a steady-state simulation. Thus, though this study confirms the possibility of running a 3D groundwater model at fine resolution over a very large territory, it is still difficult to consider its application for operational water management purpose.

Other examples include the Texas Water Development Board that has implemented several sub-models to help monitor groundwater resources at the state scale (more than 500 000 km$^2$) (Texas Water Development Board, 2018) or New Zealand where a nationwide groundwater recharge model is currently under development (Westerhoff et al., 2018).

In France, the hydrometeorological model SAFRAN-ISBA-MODCOU (Habets et al., 2008) that is used for long-term reanalyses (Vidal et al., 2010) as well as real time monitoring and forecast (Coustau et al., 2015; Singla et al., 2012; Thirel et

al., 2010) includes an explicit representation of two aquifer systems. However, the representation of these aquifer systems is rather coarse and is mostly used to have a realistic representation of the river base flow (Rousset et al., 2004) rather than providing consistent information on groundwater resources. Vergnes et al. (2012) developed a hydrogeological model dedicated to climate modelling that was first applied over France and then at the global scale (Vergnes and Decharme, 2012). However, only a single layer at the resolution of approximately 10 km over France was considered. This approach is still too

coarse to be used for groundwater management over France.

The need to have a national scale consistent representation of groundwater resources in France clearly appeared during the project EXPLORE2070 lead by the French environment ministry that aimed at providing projections of the evolution of the water resource in France including groundwater (Stollsteiner, 2012). Indeed, several regional hydrogeological models were used in this project, together with downscaled climate change projections. The results were difficult to analyse due to the

20 differences in the way the surface water balance was calculated (either lumped-parameters model or soil-vegetation-atmosphere scheme), in the initialization methods, and in the way the evolutions were estimated. Moreover, in the meantime, several regional groundwater models were developed independently by research institutions in close relationship with the stakeholders for regional water management purposes or climate impact studies (Amraoui et al., 2014; Croiset et al., 2013; Douez, 2015; Habets et al., 2010; Monteil et al., 2010; Vergnes and Habets, 2018).

In such context, the AquiFR project was built to capitalize these developments in order to provide monitoring and forecasts of groundwater resources in France, as well as long-term reanalyses and future projections. The project associates research teams in hydrogeological, numerical, and atmospheric fields. It is funded by a national stakeholder in charge of the water resource, the French Agency for Biodiversity (AFB). The main idea of AquiFR is to include existing hydrogeological models developed with different groundwater modelling software and to connect them with real time atmospheric analysis

and weather forecasts for producing relevant information for water resource management through a single numerical tool. This project also encourages new developments over areas where no groundwater models currently exist. To achieve these objectives the AquiFR hydrogeological modelling platform was developed. The main objectives of this paper are to describe this platform, to evaluate its performance against observations, and to prove its suitability and robustness for operational and research purposes.



In its present form, the AquiFR system includes 3 groundwater flow software covering 11 sedimentary aquifers and 23 karstic systems: the EauDyssée hydro(geo)logical numerical platform (Saleh et al., 2013), the MARTHE groundwater flow software (Thiéry, 2015a) and the EROS lumped model software used for karstic systems (Thiéry, 2018a). These softwares are embedded in an application developed with the OpenPALM coupling system (Duchaine et al., 2015). All these models

cover an area of about 149 000 km² and contains up to 10 overlaid aquifer layers. Prior to real time monitoring and forecast, AquiFR need to be assessed on a long-time period, which is reported on the present study. The evaluation is carried out over a 60 year period from 1958 to 2018 at a daily time step. This long-term simulation provides a unique insight on the long-term evolution of groundwater in France, as most of the groundwater data are available over about 30 years. This long-term simulation can then be used to characterize the daily situation compared to past events. The SAFRAN meteorological

reanalysis (Vidal et al., 2010), available over the French metropolitan area at an 8 km resolution, allow to supply the meteorological variables to the SURFEX land surface model (Masson et al., 2013) which evaluates the water balance over the French metropolitan area. It must be stressed that the hydrogeological models could have been classically fed with the SAFRAN reanalysis precipitation, potential evapotranspiration, and temperature data using their own water balance calculation. The combined use of SURFEX and SAFRAN provides a consistent set of hydro-meteorological data including

groundwater recharge and surface runoff from SURFEX, as well as potential evapotranspiration, precipitation, and temperature from SAFRAN. The use of these 8 km resolution fluxes made necessary the recalibration of the hydrogeological models included in the platform. More details can be found in Habets et al. (2017).

A wide range of gauging stations and piezometers were selected in order to perform the evaluation of the simulated piezometric heads, river flows and karstic spring flows. This evaluation allows to assess the performance of the platform to

identify extreme events such as groundwater floods or droughts over a long-term period.In this paper, a detailed description of the datasets and the different components of AquiFR are presented in section 2. Section 3 presents the assessment of the long-term simulation based on river flow, karstic spring flow and piezometric head observations, which is then discussed in section 4. Conclusions are drawn in section 5.

## 2    The AquiFR Hydrometeorological Modelling Platform

The AquiFR hydrogeological modelling platform was developed using the OpenPALM coupling system (Buis et al., 2005; Duchaine et al., 2015). OpenPALM allows the easy integration of high-performance computing applications in a flexible and scalable way. It was originally designed for oceanographic data assimilation algorithms, but its application domain extends to multiple scientific applications. In the framework of OpenPALM, applications are split into elementary components that can exchange data. The AquiFR platform is an OpenPALM application that currently gathers 5 components: one component

that retrieves the fluxes previously simulated by the SURFEX land surface model (Masson et al., 2013), one component for each model: the EauDyssée (Saleh et al., 2013) and MARTHE (Thiéry, 2015a) groundwater modelling software, the EROS software (Thiéry, 2018a), and a last component that synchronizes each model at each time step and gathers all the data for



post processing. All these elements are connected together and exchange data during the parallel execution of a single OpenPALM executable. Figure 1 shows the basic diagram of AquiFR as it is developed in OpenPALM. The use of OpenPALM allows to run each instance of the models in parallel over several processors. The 60 year simulation presented in this study needs approximately 1.5 days of computation time on a high-performance computer.

Prior to an AquiFR run, atmospheric forcing from an independent simulation of a numerical weather forecast model or from observed database are provided to the SURFEX land surface scheme (Masson et al., 2013). SURFEX computes the energy and surface water budget at the land-atmosphere interface and simulates surface runoff and groundwater recharge needed by the hydrogeological models in the AquiFR platform. Atmospheric forcing corresponds to precipitation, temperature, relative air humidity, wind speed and downward radiations. For the present study, the SAFRAN meteorological reanalysis is used.

The workflow of an AquiFR time step occurs in the following way. First, both the atmospheric forcing and the SURFEX groundwater recharge and surface runoff are retrieved at the beginning of the time step. Then, within the AquiFR platform, EauDyssée and MARTHE use the SURFEX fluxes to compute the simulated piezometric heads, river flows, and groundwater-surface water exchanges over regional aquifers at different spatial resolutions depending on the hydrogeological model. These groundwater-surface water exchanges include both the stream-groundwater interactions and

the overland groundwater overflows. In parallel, the EROS software computes karstic spring flows at the outlets of the karstic systems. Inputs are precipitations (snow and rainfall), temperature and potential evapotranspiration provided by the meteorological forcing inputs. Once each hydrogeological model reaches the end of the current time step, the synchronization component retrieves the current time step outputs for post-processing and allow the platform to compute the next time step.

The following subsections present a brief description of the components integrated within the OpenPALM application as well as the models currently included in AquiFR.

**2.1 The SAFRAN meteorological reanalysis**

The SAFRAN analysis system is a mesoscale atmospheric analysis system for surface variables. It provides meteorological forcing data over France on an 8 by 8 km grid at the hourly time step using observed data and atmospheric simulations.

Originally intended for mountainous areas, it was later extended to cover France (Quintana-Seguí et al., 2008). SAFRAN analyses eight variables: the rainfall, snowfall, incoming solar and atmospheric radiations, cloudiness, 2 m air temperature, 2 m relative humidity and 10 m wind speed.SAFRAN is based on climatic zones where the atmospheric variables only vary according to the topography. More than 600 homogeneous climate zones are defined over France. The average area for each zone is about 1000 km$^2$ so that each one contains one surface meteorological station and at least two rain gauges SAFRAN

uses all the observations available to analyse each atmospheric variable except for radiations. For each variable, values are assign to given altitudes using an optimal interpolation method. The analyses are computed every 6 hours and an interpolation is made to an hourly time step and radiation fluxes are computed using a radiative transfer scheme. The daily precipitation rates are estimated using a wide range of daily rain gauges and converted to hourly data using the evolution of



the air relative humidity. The vertical profiles of the atmospheric parameters are then computed in each climatic zone and the values are spatially interpolated over the 8 km grid as a function of the altitude within each climatic zone.Further details on the SAFRAN analysis system can be found in Quintana-Seguí et al. (2008).

## 2.2 The SURFEX modelling platform

SURFEX is a modelling platform aiming to simulate the water and energy fluxes at the interface between the surface and the atmosphere (Masson et al., 2013). SURFEX is built to be coupled to forecast and climate models and then can be used over different spatial and temporal scales. SURFEX gathers several physical schemes in a single platform, allowing the simulation of the urban surfaces and the main components of the water cycle: sea and ocean, lake, vegetation and soil. In the present study, SURFEX is used in offline mode in order to provide groundwater recharge and surface runoff to the AquiFR
platform.

Land surface processes are taken into account using the Interaction between Soil Biosphere and Atmosphere (ISBA) land surface scheme. ISBA uses a short list of parameters depending on vegetation and soil types. The tempral evolution of the soil water and energy budget is computed using a multilayer soil scheme based on the explicit resolution of the one-dimension Fourier law as well as the mixed form of the Richards equation. (Boone et al., 2000; Decharme et al., 2013).
Surface/groundwater capillary exchanges can be explicitly taken into account (Vergnes et al., 2014) as well as the vertical root profile in the soil (Braud et al., 2005). Further details on ISBA can be found in Decharme et al. (2013).

## 2.3 The EauDyssée groundwater modelling software

The EauDyssée modelling platform gathersnumerical modules representing several hydrological processes, the most important being the aquifer module based on the Simulation of Multilayer Aquifers (SAM) regional groundwater model
(Ledoux et al., 1989) and the riverrouting scheme based on the Routing Application for Parallel Computation of Discharge (RAPID) model (David et al., 2011).

The SAM model computes the evolution of the piezometric heads of multilayer aquifers using a finite difference numerical scheme to solve the groundwater diffusivity equation with a square grid discretization. Groundwater horizontal flows are two-dimensionals while vertical flows through aquitards are represented by leakage.Therefore, unconfined and confined
aquifers can be represented. SAM was successfully used to predict groundwater and surface water flows in different basins of various scales and hydrogeological contexts: the Seine basin (Viennot, 2009), the Somme basin (Habets et al., 2010), the Loire basin (Monteil et al., 2010) or the Rhine basin (Thierion et al., 2012; Vergnes and Habets, 2018).

The RAPID software is a river routing model based on the Muskingum routing scheme (David et al., 2011). It can be coupledto groundwater and land surface models. Volume and river flow are computed along a river network discretized into
square grid-cells to ease the simulation of the exchange with groundwater.



## 2.4 The MARTHE groundwater modelling software

The MARTHE (Modelling Aquifers with Rectangular cells, Transport and Hydrodynamics) computer code is the hydrogeological model from the French Geological Survey (BRGM) (Thiéry, 2015a, 2015c, 2015b). MARTHE embeds single to multilayer aquifers, hydrographic networks and the exchanges with the atmosphere (rainfall, snow and

evapotranspiration) for the computation of the soil water balance. It is designed for 2D or 3D modelling of flows and mass transfers in aquifer systems, including climatic, human influences and possible geochemical reactions. Groundwater flow is computed by a 3-D finite volume approach to solve the hydrodynamic equation based on the Darcy's law and mass conservation, using irregular rectangular grids, with the possibility of nested grids. Based on a kinematic wave approach, flow in river networks is fully coupled to groundwater flow. Other options are available and can be integrated to the

simulation: mass transfer for pollutants in water, temperature effects, impact of salinity, degradation of pollutants, transfers in the unsaturated zone and geochemical reactions.

This software has been widely used for groundwater resources management in France: for example in the Somme River basin (Amraoui et al., 2014), in the Poitou-Charentes region (Douez, 2015), in the Basse-Normandie region (Croiset et al., 2013) or in the Aquitaine sedimentary basin (Saltel et al., 2016). It has also been used in other environmental fields such as

pollutant infiltration in unsaturated zones (Herbst et al., 2005; Thiéry et al., 2018) or for the simulation of pollution plume coming from a contaminated area.

## 2.5 The EROS software

The EROS (set of rivers organized in sub-basins) numerical code is a hydro-climatic rainfall-river flow-piezometric head distributed model dedicated to large river systems (Thiéry, 2018a; Thiéry and Moutzopoulos, 1992). It allows the simulation

of river flow or karstic spring flow and piezometric head in heterogeneous river basins. These river basins are described in EROS as a cluster of elementary lumped-parameter hydrological models connected with each other. For each sub-model, a hydroclimatic lumped model allows to compute the local river discharge at the outlet of the sub-model and the piezometric head in the underlying water table. Each sub-model simulates the main mechanisms of the water cycle through simplified physical laws (Thiéry, 2015d). Snow accumulation, snow melting and pumping is taken into account. The total river flow at

the outlet of each sub-basin is computed from the upstream tree of sub-basins.

EROS was initially developed to simulate regional watershed avoiding the complexity of a more complex spatially physically based model without sacrificing the performance. In the framework of the AquiFR project, this software was used and adapted in order to simulate in a single run 23 karstic systems as independent sub-models, making the simulation of these karstic systems less expensive in terms of computational burden with respect to the use of 23 distinctive models

(Thiéry, 2018b).



## 2.6 The regional models implemented in the AquiFR platform

AquiFR aims at covering all groundwater resources in France. Figure 2 shows the main aquifers covering France classified by geological type as defined in the French hydrogeological reference system BDLISA (https://bdlisa.eaufrance.fr/). The current version of AquiFR gathers 13 spatially distributed models corresponding to regional single or multilayer aquifers

(Table 1 and Figure 3). These hydrogeological models were developed independently most often based on stakeholder requests. The water budgets in these models were usually computed using less physical methods and atmospheric local data, precipitation and temperature, that differ from the physically-based approach using SURFEX and SAFRAN. As a result, in order to be consistent with the estimation of the groundwater recharge estimated by SURFEX, each regional model was recalibrated based on the SURFEX fluxes (Habets et al., 2017). Generally speaking, observations and periods of calibration

were the same as those initially used to develop each model. Hydrodynamic parameters, including hydraulic conductivities and specific yields, were modified based on hydrogeological expertise in order to obtain the best fit between simulated and observed river flow and piezometric heads. It should be mentioned that for the MARTHE Somme basin model, in order to get an accurate simulation, at each time step, the choice was made to divide the total runoff provided by SURFEX into surface runoff and groundwater recharge using the GARDENIA water balance scheme inside the MARTHE computer code.

Some regions are simulated by two spatialized models (Figure 3): the Somme and the Basse-Normandie basins are covered by MARTHE and EauDyssée models, and the chalk aquifer of the Seine basin is covered by both the EauDyssée Seine model and four EauDyssée sub-models (Marne-Loing, Marne-Oise, Seine-Eure, and Seine-Oise regional models, see Figure 3). This allows a multi-model approach, which can be useful for forecast and climate change impact studies. For these regions, the results presented in this paper correspond to the models that were considered as the best calibrated against the

SURFEX fluxes. It corresponds to the four EauDyssée sub-models over the Seine basin and the MARTHE models for the Somme and Basse-Normandie regions. Figure 3 also shows the 23 karstic systems (median catchment area of 99 km$^2$) simulated by EROS. (Thiéry, 2018b) as well as the hard rock aquifer in Britany that will be simulated using a hillslope model (Courtois, 2018; Marçais et al., 2017) and integrated in the near future.

Groundwater withdrawals are integrated as input data in the spatially distributed models. On annual average and with respect

to the total surface area of the simulated domain, it corresponds to about 16 mm/year (that is about distributed in more than 16 000 grid cells. Data on groundwater pumping are provided by the regional water agencies. The quality of the data set as well as its temporal extension varied for each regional modelling, although the latter does not exceed 20 years. Further details on regional models can be found in the references listed in Table 1. To extend the pumping estimation to the 1958-2018 period, a monthly mean annual cycle is used for the years without data.

Some spatialized models also need to prescribe time dependant boundary conditions. For the Alsace and the Poitou-Charentes models it is required to provide upstream river flow from river flow observations. If the observed data don't cover the full period, the daily mean annual cycles are calculated from the observed period and applied to the simulated period not



covered by observed data. In the near future, a new method based on a lumped-parameter rainfall-runoff model integrated in the MARTHE computer code will be implemented in order to better estimate these upstream flows.

## 3 Results

The long-term simulation was carried out over a 60 year period from the August 1, 1958 to July 31, 2018 at a daily time step
using the SAFRAN meteorological reanalyses. The mean precipitations corresponding to the simulated domain of Figure 3 and averaged over the 60 year period is equal to 743 mm/year. The surface water budget is then computed by SURFEX from the SAFRAN outputs. The mean simulated effective rainfall is partitioned between 163 mm/year of groundwater recharge and 60.5 mm/year of surface runoff.

The evaluation of this simulation was made using the numerous in situ datasets available in France. Observed piezometric
heads over France are available in the "Accès aux Données sur les Eaux Souterraines" (ADES) database (http://www.ades.eaufrance.fr/). 639 observation boreholes covering the regional models included in AquiFR were selected, corresponding to both confined and unconfined aquifers, and with at least 10 years of continuous time series. Figure 4 shows the temporal evolution of the number of daily measurements along the 60 year period. Starting in 1958, only a few measures are available. Starting from 1970, the number of wells increases slowly to reach about 100 in 1990. Then the number of daily
measurements quickly increases to reach more than 450 in 2010. This number remains stable then, except for the last year (2018) with a decrease because the datasets were not yet fully transmitted. In situ daily river flow observationsat 228 gauging stations were also selected for evaluating the daily simulated river flows from the Hydro database (http://hydro.eaufrance.fr/).

### 3.1 Piezometric head

In order to evaluate the quality of the simulation, two statistical critera are used: the bias and a normalised Root Mean Square Error (RMSE) bias-excluded (NRMSE-BE). Figure 5a shows the spatial distribution of the bias scores for the 639 observed piezometers selected over the simulated regional multilayer aquifers. A positive (negative) value means that the simulation overestimates (underestimates) the mean piezometric head with respect to the observation. The north of the Loire river basin, corresponding to the Beauce region, shows a significant underestimation of the mean observed groundwater level.
Elsewhere, no significant patterns appear. Figure 5b summarizes these results with the accumulated distribution of the absolute biases for all the piezometers. 40% and 60% of the absolute biases are lower than 2 m and 4 m respectively.

The computation of the RMSE scores is very affected by these biases. Therefore, we compute a RMSE bias-excluded score in order to avoid these biases and to better assess the simulation in terms of amplitude and synchronization. Moreover, this RMSE bias-excluded score is normed with respect to the observed standard deviation for each observation. It allows to take
into account the differences of variability between the observed points and to better compare them with each other. Figure 6a shows the spatial distribution of these normalised RMSE bias-excluded (NRMSE-BE) scores and Figure 6b summarizes



these results with the accumulated distribution of NRMSE-BE for all the piezometers. 16% and 61% of these scores are lower than 0.6 and 1 respectively, while 88% of them are lower than 2. Some piezometers that were affected by important biases in Figure 5a exhibit however good NRMSE-BE scores, in particular over the Loire river basin and in the northern Poitou-Charentes region.

Five examples of simulated and observed daily evolution of piezometric heads are shown in Figure 7. These piezometers are encircled in Figure 2 and statistical scores are available in Table 2. They were chosen to characterize different hydrogeological contexts. The first piezometer named Omiécourt is located in the Chalk aquifer of the Somme River basin. The temporal evolution of the groundwater level is characterized by multiyear cycles well captured by the model. However, the simulation displays annual cycles that are not observed. These annual cycles explain why the NRMSE-BE score is equal

to 0.81 while the bias score is equal to -0.15 m. The two piezometers named Ruffec and Le Bec hellouin correspond to limestone aquifers and are located in the Poitou-Charente region and near the coast of the English Channel respectively. The first one is characterized by large annual cycles with wide amplitudes. The model is able to reproduce these annual cycles (correlation of 0.82) but with an underestimation of the peaks leading to a negative bias score of -1.44 m. The Le Bec hellouin piezometer is characterized by both multiyear and annual cycles that are captured by the model except between the

2005 and 2015 years where the simulated groundwater level is underestimated with respect to the observation. The piezometer named Farceaux is located in a chalk aquifer in the Seine River basin. It is characterized by a systematic bias of about -8.3 m. Otherwise, the multiyear and annual cycles are well reproduced by the model, which is confirmed by the NRMSE-BE score equal to 0.52. The last example corresponds to a piezometer for which the model cannot reproduce the strong seasonal decrease of level occurring each year. Such behaviours in the observation are likely due to groundwater

withdrawals that are not well prescribed in the model near this observation well.

## 3.2 The Standardized Piezometric Level Index

One way to evaluate the ability of the simulation to capture extreme events is to use the Standardized Piezometric Level Index (SPLI). The SPLI is an indicator used to compare groundwater level time series and to characterize the severity of extreme events such as long dry period or groundwater overflows (Seguin, 2015). Assessing the ability of the AquiFR

modelling platform to reproduce this indicator is important since the main objective of this platform is to predict such extreme events in short-to-long terms hydrogeological forecasts for groundwater management. The SPLI indicator is based on the same principles as the Standardised Precipitation Index (SPI) defined by (McKee et al., 1993) to characterize meteorological drought at several time scales. The SPLI values most often range from -3 (extremely low groundwater levels corresponding to a return period of 740 years) to +3 (extremely high groundwater levels). The SPLI is normalized so that

wetter and drier periods can be represented in a similar way all over the French national territory. The SPLI is currently used in France for the Monthly Hydrological Survey (MHS) (Office International de l'Eau, 2019). This MHS provides monthly information to the policymakers and the public on the hydrological state of groundwater. The SPLI is categorized into seven classes summarized in Table 3 from the driest to the wettest conditions. According to Seguin and Klinka (2016), a set of



piezometers were chosen in order to compute the SPLI indicator in this MHS with the following characteristics: a continuous time series with at least 15 years and no impact of pumping wells. Among the 639 selected observation wells in Figure 5 and Figure 6, 103 contribute to the calculation of the SPLI in the MHS.

The correlation and the Nash-Sutcliffe model efficiency score (NSE) (Nash and Sutcliffe, 1970) are used to evaluate the
SPLI indicator. A negative NSE means that the mean observed signal is a better predictor than the model. A NSE above 0.7 is generally accepted as a reasonable estimate of the signal dynamic, however depending on the hydrogeological and climate context of the basin. It is often applied to compare observed and simulated river flows but can be used for other variables such as the SPLI indicator. It sensitivity to high-frequency fluctuations makes its use for comparing groundwater levels less obvious. Figure 8 and Figure 9 show the spatial distribution of the NSE criterion and the correlation scores respectively
computed between the observed and simulated SPLI indicator for the 103 selected piezometers of the groundwater domain. 20% of the NSE scores are greater than 0.7 and 55% greater than 0.5, while 11% are lower than zero. In parallel, 64% of the correlation scores are greater than 0.7.

The left part of Figure 10 focuses on five examples of observed and simulated temporal evolutions of the SPLI indicator. These piezometers correspond to the ones shown in Figure 7 and are part of the selected piezometers used for the MSH.
Background colours correspond to the classification of the SPLI from the driest (red) to the wettest (blue) hydrological conditions as shown in Table 3. Table 2 presents the related NSE and correlation scores. The NSE scores computed for these SPLI time series are all greater than or equal to 0.6 except for the Bourdet piezometer characterized by a NSE score equal to -0.51. This lower score may be due to a lack in the inputs of the model, such as the absence of withdrawal data in its vicinity. The SPLI is a statistical calculation with the particularity to centre the temporal evolution of the piezometric head on zero,
hence removing the potential biases between the observed and simulated groundwater levels. This is the reason why the systematic biases found in Figure 7 do not appear in the monthly SPLI evolutions in Figure 10, in particular for the Farceaux piezometer. Moreover, the SPLI indicator normalized the amplitude of both the observed and simulated temporal evolution of groundwater levels. The right part of Figure 10 describes the histograms in percentage of the simulated (in blue) and observed (in black) monthly SPLI values distributed against the classes of Table 3. This classification is similar for both the
observed and simulated SPLI of Ruffec and Le Bec Hellouin piezometers. The occurrences of the wetter conditions are well reproduced for Omiécourt piezometer but the model tends to overestimate the importance of the driest conditions when compared to the observations (26% and 18% of "moderately dry" events for the simulation and the observation respectively and 4.5% and 7.2% of "very dry" events for the simulation and the observation respectively). For the Farceaux piezometer, the model underestimates the occurrences of the driest events and overestimates the occurrences of the wetter events..
Despite the poor scores obtained for the Bourdet piezometer, in particular for the correlation scores, the distribution of all the monthly SPLI values with respect to the classes of Table 4 is similar for both the observation and the simulation.

The MHS published every month in France for water resources management includes the calculation of the SPLI. As an example, Figure 11a shows the observed SPLI values calculated for the 103 selected piezometers for June 2016. We chose this specific month since it follows large precipitation events that leads to floods in the Seine and Loire basins (Philip et al.,





2018). Figure 11b shows the simulated SPLI values computed for this specific month. The model reproduces the overall pattern of normal and wet conditions but tends to overestimate the importance of the moderately wet conditions: 17% (29%) of the simulated (observed) piezometers are in normal conditions, 50% (32%) are in moderately wet conditions, 14% (12%) are in wet conditions, and 16% (18%) are in extremely wet conditions. The background map of Figure 11b show the SPLI

computed in the cells of the whole outcropping domain. These SPLI values were computed with respect to a 30 years reference period from 1981 to 2010, which might lead to differences between the simulated SPLI map and observed values. This map shows a large area of extremely wet conditions located in the south of the Loire River, which refers to the extreme event episode of rainfall from the end of May 2016.

### 3.3 River flow and karstic spring flow

The 23 karstic systems simulated by the EROS model are evaluated against gauging stations located at the outlet of the corresponding karstic systems. All these gauging stations were also used to calibrate the model (Thiéry, 2018b). Figure 12 compares the observed and simulated monthly river flows for four examples of karstic systems located in Figure 2. There is a tight agreement between the observation and the simulation. The NSE scores of the square root of the daily river flows are given for each example. Using the square root of the daily river flow allows to attenuate the importance of the flood peaks

characterizing these small karstic systems and enables to better evaluate the river spring flow simulation. Such transformation is necessary because of the excessive sensitivity of the NSE criteria to extreme values in a river flow time series (Legates and McCabe Jr, 1999). For these four examples, all the NSE scores are greater than 0.7.

The distributed models included in the AquiFR modelling platform integrate river networks and the simulation of river flows on each river grid cell. 228 gauging stations were selected to evaluate the simulated river flows. These gauging stations

correspond to the ones that were used for calibrating and evaluating each model (see references in Table 1). They were kept to evaluate the AquiFR platform (Habets et al., 2017). Figure 13 compares the observed and simulated daily river discharges for four of them: the Charente River, the Somme River, the Seine River and the Loire River. The locations of these gauging stations is shown in Figure 2. The Charente River and the Somme River correspond to watersheds with areas lower than 10 000 km$^2$ and the Loire River and the Seine Rivers to regional watersheds with areas greater than 80 000 km$^2$. Three

statistical scores were selected to evaluate the performance of the models: the NSE, the correlation and the annual discharge ratio criterion (Ratio = $Q_{sim}/Q_{obs}$). These statistical scores are summarized in Table 4 for the gauging stations of Figure 13. The simulated river flow of the Charente River underestimates the peak floods, which leads to a ratio score of 0.81. The river flows of the Seine River and the Loire River are well reproduced with daily NSE scores of 0.86 and 0.9 respectively. The Somme river flow is also well reproduced with a NSE score equal to 0.69 due to a lower ratio score of 0.92.

Figure 14a shows the spatial distribution of NSE scores for the 228 gauging stations helping to evaluate the distributed hydrogeological models (circles) and the 23 karstic systems (stars). NSE scores calculated for the karstic systems correspond to the square root of the daily river flows. The corresponding accumulated distributions are shown for the 23 karstic systems and for the 228 gauging stations in Figure 14b and Figure 14c respectively. 80% of the NSE score using the square root of




the daily river flows are greater than 0.8. Regarding the results of Figure 14c, for rivers in continuous aquifers, 27% of the NSE scores are greater than 0.7, 58% are greater than 0.5, and 22% are negatives.

## 5 Discussion

The results obtained over the 1958-2018 period demonstrates the feasibility and the utility to gather several regional models developed separately in several research institutes into a single numerical tool to provide simulations of the water resource at a daily time step at the national scale. It was shown that the AquiFR platform is able to reproduce the evolution of the observed hydrological variables, including piezometric levels and river flows, with reasonable statistical scores. Some regions are nevertheless better reproduced than others. For example, the Loire and the Nord-Pas-de-Calais regions exhibit a poor NRMSE-BE scores in Figure 5 than the other regions. Part of the error is linked to the estimation of the groundwater recharge by SURFEX essentially because the models were developed and calibrated independently using various methods and data to compute groundwater recharge (see references in Table 1). For consistency, the development of AquiFR reinforces the need to calibrate these models based on the SURFEX forcing fields. This work was accomplished for most of the models included in AquiFR except for some of them, including the Loire River basin (Habets et al., 2017). The use of an inverse model as the one proposed by Hassane Maina et al. (2017) should help to improve such calibration.

Starting from scratch with an integrated method could have prevented the burden of maintaining each model separately and handling the different outputs of the models. Such method was applied by Kollet et al. (2018) over the North Rhine-Westphalia domain using Parflow-CLM by integrating all the physical processes related to groundwater/surface water into a single numerical tool. De Lange et al. (2014) used an approach closer to AquiFR by coupling five physically different models for different water domains with different concepts, different temporal and spatial scales, and different national and regional databases altogether embedded into the National Hydrological Instrument NHI. These two models are used for integrated water management and policymaking issues. The areas covered by these models are 22 500 km$^2$ and 41 500 km$^2$ respectively. According to Figure 2, the BDLISA database references regional sedimentary aquifer systems (in green) and alluvial aquifers (in blue) in France both covering an area of about 355 000 km$^2$. Reaching the complexity of a fine-tuned regional model, including the geometrical, geological and physical contexts, in a single integrated numerical tool covering such large territory would be time consuming to build, to calibrate, and to evaluate. Itwould also necessitate big resources in computational power. An attempt was made by Vergnes et al. (2012) to simulate in a single integrated model groundwater over the French metropolitan territory. Even though the results obtained were good enough to be used for large-scale climate applications, it was not fitted for operational water management: only one layer was defined at the coarse resolution of about 10 km, no pumping were defined, no calibration was achieved and very simplified parameterizations were used.

To overcome this difficulty, the choice was made for AquiFR to bring together different models developed independently. Currently, the area covered by the platform is equal to around 133 000 km$^2$ with a number of layers that can reach 10 layers for some models, and these numbers will increase in the future with the addition of new models. This multi-model approach





allows to promote the share of knowledge in hydrogeology and to gather the competencies accumulated in the different research institutes involved in AquiFR in water resource modelling. Moreover, thanks to the evolutive approach of the OpenPALM coupling software, the platform facilitates the addition of new software and new models.

Results from AquiFR show a global view of the performance of the AquiFR platform but are characterized by uncertainties. These uncertainties are mainly related to the calibration of the models and to the lack of some input data like groundwater abstractions. Indeed, the regional models have been calibrated over a shorter period compared to the long-term simulation. Consequently, they do not take into account in the calibration process the extreme climatic conditions known over the 1958-2018 period. Other uncertainties may be related to the choices of the resolution, the databases used, the geometry of the models, and more generally the representation of the physical processes in the hydrogeological software. Some regions are better monitored than others and the global view of the performance of the AquiFR platform is certainly affected by this. Moreover, the chosen method of evaluation based on statistical scores such as NSE could also be improved. Indeed, some authors report that the use of more realistic upper and lower benchmarks for each simulated basins could improve the judgement of model performances with respect to the climate and hydrogeological context of the basins (Pappenberger et al., 2015; Seibert et al., 2018). In order to diminish these uncertainties, a long-term calibration effort using a denser observation network should be undertaken to improve the AquiFR performance.

The AquiFR platform can be seen as an improvement of the SIM hydrometeorological tool for operational water management purpose (Habets et al., 2008). These two systems share common points as the SURFEX land surface model or the groundwater component of the EauDyssée platform. However, the SIM tool uses coarse hydrogeological modelling and mainly focuses on operational forecasts of river flows. The AquiFR platform is intended to focus also on the forecast of groundwater levels for the multilayer aquifers and karstic systems described in Figure 3. To achieve this goal, the SPLI indicator will be calculated to provide forecasts of extreme events. For this purpose, AquiFR is able to produce different representation of this indicator and to compare it with other variables depending on the need of the stakeholders. For example, Figure 15 compares the simulated time series of daily mean groundwater recharge, stream-groundwater exchanges budget,monthly mean piezometric head and SPLI averaged over the chalk aquifer of the Somme model. It gives a global view of the past states of groundwater related to climatic extreme events, such as the severe flood of the 2001 year, characterized by groundwater flooding and sustained stream-to-groundwater exchanges (Amraoui and Seguin, 2012).

## 6 Conclusions

This study introduces the AquiFR hydrogeological modelling platform aiming to provide at the French national scale short-term to seasonal hydrological forecast as well as real-time monitoring for daily water management and long-term simulations for climate impact studies. It was developed using a coupling software in order to gather inside the same numerical tool different softwares for different water domains and several models covering a set of French multilayer aquifers. Daily surface runoff and groundwater recharge prrovided by the SURFEX land surface model forced by the SAFRAN





meteorological analysis were used to feed AquiFR for simulating the daily evolution of groundwater levels and river flows of French regional multilayer aquifers and karstic systems over the 1958-2018 period.

The results confirm the feasibility of gathering independent hydrogeological model developed in different research institutes into the same coupling platform. All these models were initially developed and calibrated on shorter period with
heterogeneous geological and meteorological databases. Some of these models were recalibrated against the SURFEX and SAFRAN fluxes. The evaluation of the 1958-2018 long-term simulation shows a good comparison with the observations available on the same period. It confirms the relevance of using AquiFR as a tool for future long-term impact studies. The evaluation of the SPLI indicator also shows that AquiFR could be used in an operational context for predicting future extreme events in a similar way than the MSH produces each month in France to quantify the hydrological state of the
aquifers, provided the necessary caution in terms of communication of model uncertainties and performances.

The advantage of this platform lies in its modularity. AquiFR encourages the development of groundwater modelling where it is missing and, more generally, it has the potential to be a valuable tool for many applications in water resource management and research studies. In the future, more regional spatial models developed with MARTHE or EauDyssée will be included in order to extend the cover of AquiFR, as the Tarn and Garonne aquifer (Figure 3). A new software will be
included in order to simulate bedrock aquifers located in Britany (see Figure 3) (Courtois, 2018). A new modelling method based on a lumped-parameter rainfall-runoff model will be used to provide upstream river flows as boundary conditions for the MARTHE model that required it. Another project will be to add a river network covering all the French territory. Assessment of the seasonal forecast of the groundwater resource is now on progress (Roux, 2018). Since errors in the initial conditions can alter significantly the skill of the forecast, dedicated studies on data assimilation to improve initial state
conditions are also done in parallel (Hassane Maina et al., 2017).

**Acknowledgements.**

The authors are grateful for financial support from the French Agency for Biodiversity (AFB) and the French Ministry of Ecological and Solidarity Transition. The authors are also grateful to the research institutions that provide computer access, data, models and software for making this project feasible (BRGM, CERFACS, Météo France, Mines ParisTech,
Geosciences Rennes). Special thanks come to Claire Magand, Thimotée Leurent and Bénédicte Augeard for supporting this project with AFB, and Nathalie Dörfliger who supports this project with BRGM. Finally, the authors wish also to thank Eric Martin and Jean-Michel Soubeyroux that were involved at the beginning of this project.

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




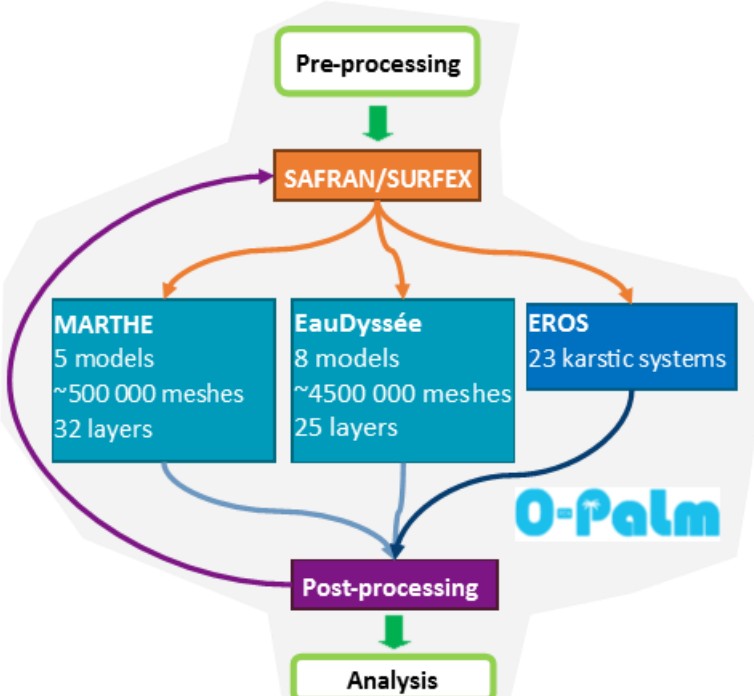

**Figure 1: Scheme of the AquiFR coupling platform.**



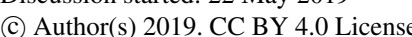

**Figure 2: Main aquifers of France classified by geological type from the BDLISA database (https://bdlisa.eaufrance.fr/). The names of the gauging stations and piezometers shown in Figure 7, Figure 12 and Figure 13 are written.**



**Figure 3: Map of the regional multilayer aquifers and the karstic systems simulated in AquiFR. The outlines of the models are also shown with colours corresponding to the outcropping aquifers with respect to their geological contexts. Grey areas correspond to models that will be integrated in a near future.**





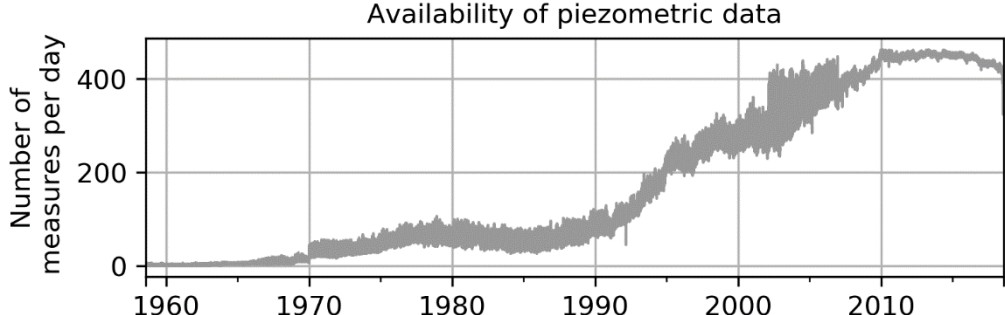

**Figure 4: Temporal evolution of the number of piezometric head measurements per day among the 639 selected piezometers over the 1958-2018 simulated period.**





Figure 5: (a) Spatial distribution of the bias scores calculated between the simulated and observed piezometric heads for the 639
selected piezometers. The grey background colour corresponds to the simulated aquifer domain. (b) Accumulated distribution of
absolute bias scores for all the piezometers.



**Figure 6: (a) Spatial distribution of NRMSE-BE scores calculated between the simulated and observed piezometric heads for the 639 selected piezometers. The grey background colour corresponds to the simulated aquifer domain. (b) Accumulated distribution of NRMSE-BE scores for all the piezometers.**





**Figure 7: Daily observed (dotted blue) and simulated (red) piezometric head variations for the five piezometers encircled in green in Figure 2.**





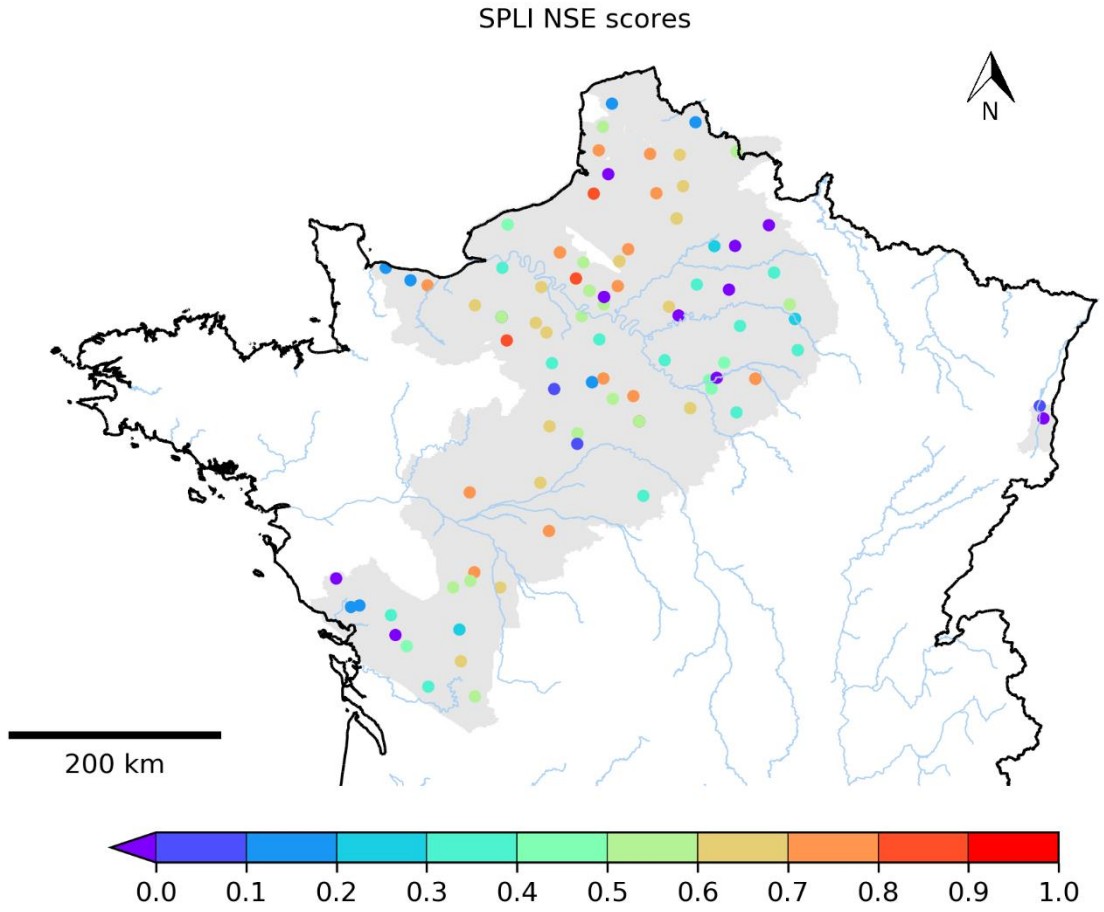

**Figure 8: NSE criterion calculated between the observed and simulated SPLI for the 103 selected piezometers.**





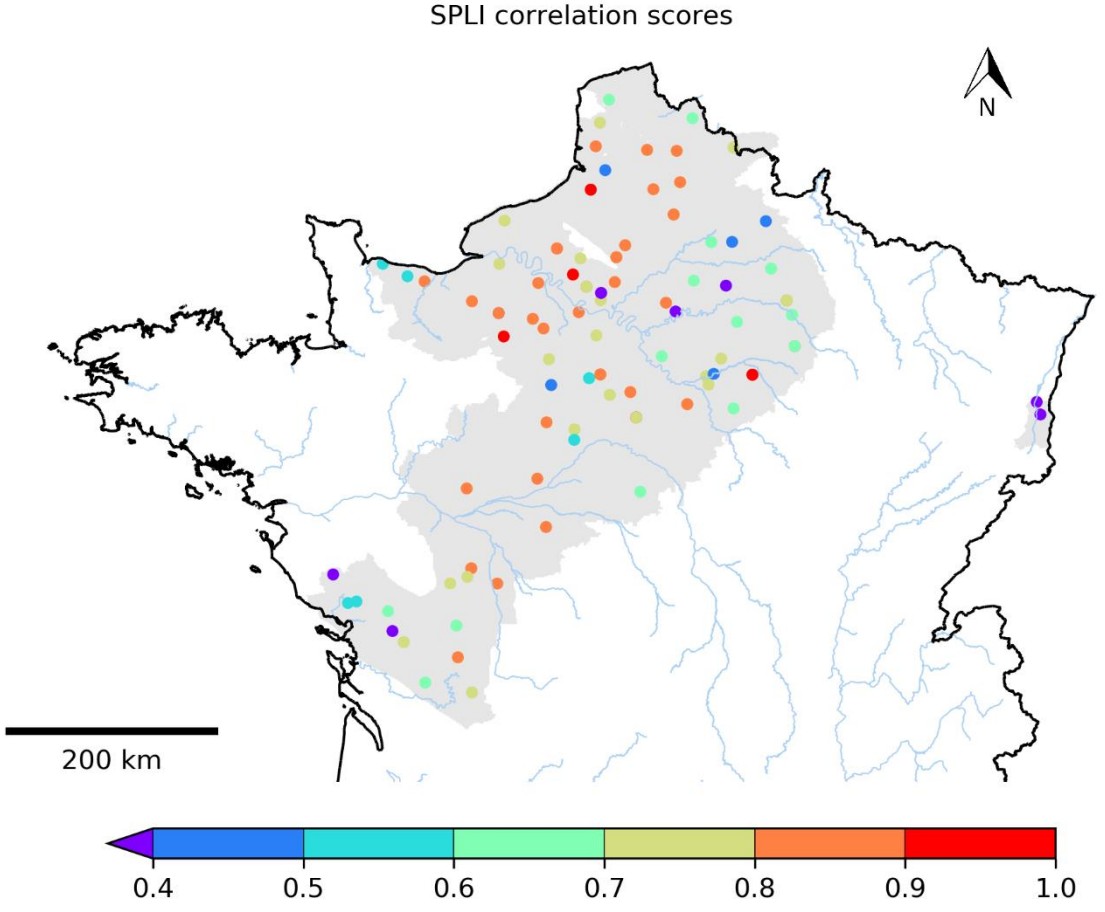

**Figure 9: Correlation scores calculated between the observed and simulated SPLI for the 103 selected piezometers.**





Figure 10: (left) Monthly observed (dotted blue) and simulated (red) SPLI indicator variations for the five piezometers encircled in green in Figure 2. Font colours correspond to the classes of Table 3 from the driest (red) to the wettest (blue) intervals. (Right) Histograms in percentage of the SPLI values distributed against the classes of Table 3.



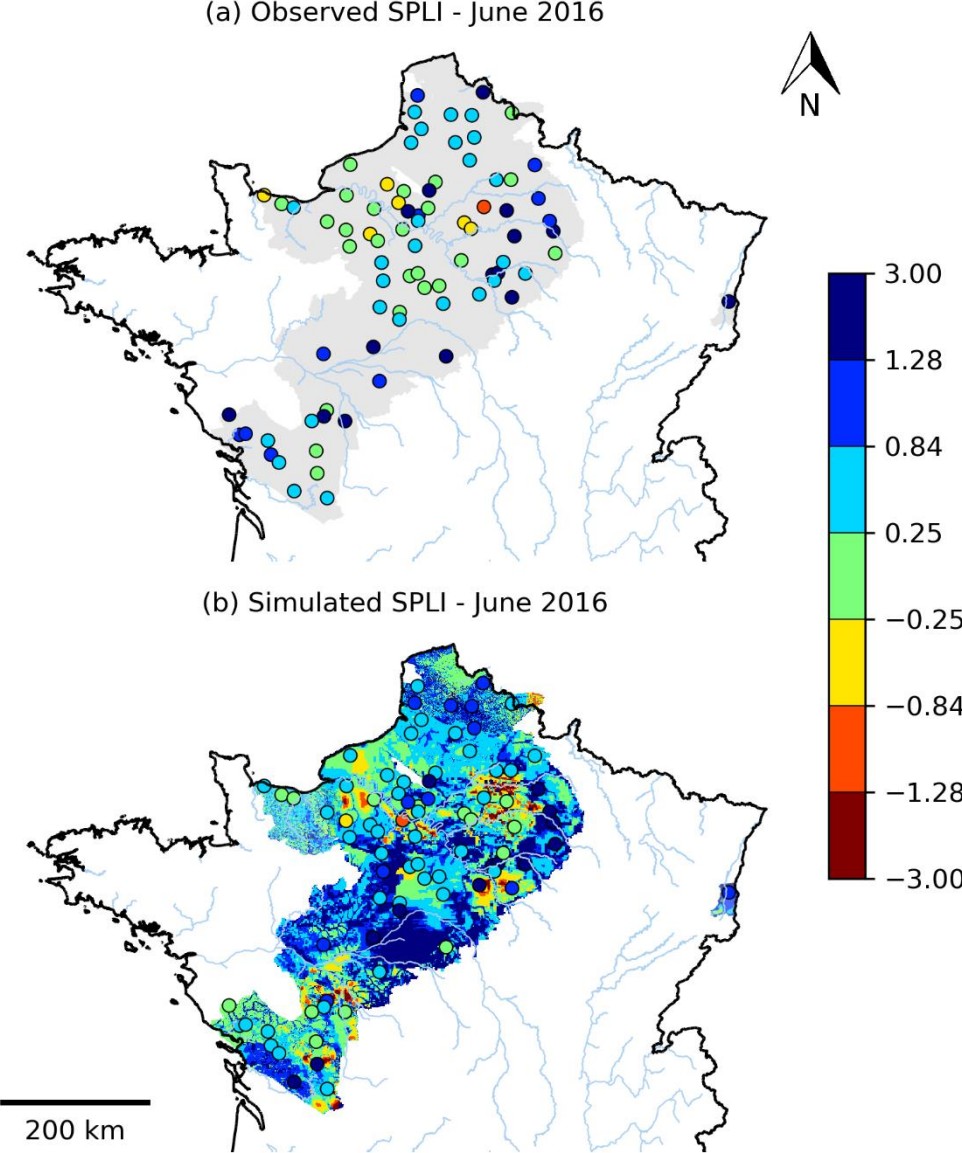

**Figure 11: SPLI indicators calculated for the august 2016 month for the (a) observed and (b) simulated piezometers. (b) The SPLI indicators calculated for all the piezometric heads of the aquifer grid are also shown.**



![Figure 12: Four time series plots comparing observed (dotted blue) and simulated (red) river flows for Fontaine de Vaucluse (NSE = 0.92, Drainage area = 1600 km²), Lison Nans sous Sainte Anne (NSE = 0.92, Drainage area = 180 km²), Fontestorbes (NSE = 0.81, Drainage area = 120 km²), and Cent Fonts (NSE = 0.9, Drainage area = 100 km²).]

**Figure 12: Monthly observed (dotted blue) and simulated (red) river flows of the gauging stations monitoring the four karstic systems encircled in red in Figure 2. NSE scores for the square root of the daily river flows and drainage area are given in parenthesis for each gauging station.**





**Figure 13: Daily observed (dotted blue) and simulated (red) river flows for the four gauging stations encircled in yellow in Figure 2.**



Figure 14: (a) Spatial distribution of the NSE scores calculated between the observed and simulated karstic spring flows and river flows for the gauging stations monitoring the 23 simulated karstic systems (stars) and the 228 selected gauging stations located in the distributed hydrogeological models (circles). Accumulated distribution of NSE scores for (a) the 23 karstic systems and (b) the 228 gauging stations of the distributed models are also shown. NSE scores for karstic systems are computed using the square root of the daily karstic spring flows. The simulated river network is shown in the background.





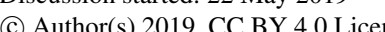

Figure 15 : (a) Spatial average over the Somme model of daily mean groundwater recharge, (b) daily mean stream-groundwater exchanges budget over the Some river network, (c) spatial average over the Somme model of the monthly mean piezometric head and (d) monthly SPLI. Red colors in (b) indicate groundwater to river flows and blue colors stream to groundwater flows. The background colors in (d) correspond to the SPLI classes from Table 3.





**Table 1: Short description of the regional multilayer aquifer models available in AquiFR**

| Software | Model | Number of layers | Number of cells | References |
|---|---|---|---|---|
| EauDyssée | Basse-Normandie | 4 | 37 667 | (Thierion, 2007) |
| | Loire | 3 | 37 620 | (Monteil et al., 2010) |
| | Marne-Loing | 4 | 66 235 | (Viennot and Abasq, 2013) |
| | Marne-Oise | 2 | 45 904 | (Viennot and Abasq, 2013) |
| | Seine | 6 | 41 609 | (Viennot, 2009) |
| | Seine-Eure | 1 | 57 306 | (Viennot and Abasq, 2013) |
| | Seine-Oise | 4 | 87 178 | (Viennot and Abasq, 2013) |
| | Somme | 1 | 63 226 | (Korkmaz, 2007) |
| MARTHE | Alsace | 3 | 40 947 | (Noyer and Elsass, 2006) |
| | Basse-Normandie | 10 | 93 800 | (Croiset et al., 2013) |
| | Nord Pas-de-Calais | 10 | 226 077 | (Bessière et al., 2015) |
| | Poitou-Charentes | 8 | 90 084 | (Douez, 2015) |
| | Somme | 1 | 66 924 | (Amraoui et al., 2014) |

**Table 2: Statistical scores of the comparison between the simulated and observed daily evolution of the piezometers shown in Figure 7.**

| Piezometer | Model | Time series | | | SPLI | |
|---|---|---|---|---|---|---|
| | | NRMSE-BE | Correlation | Biases (m) | NSE | Correlation |
| Omiécourt | Somme | 0.81 | 0.77 | -0.15 | 0.65 | 0.83 |
| Ruffec | Poitou-Charentes | 0.58 | 0.82 | -1.44 | 0.6 | 0.79 |
| Le Bec Hellouin | Basse-Normandie | 0.57 | 0.84 | -2.76 | 0.73 | 0.86 |
| Farceaux | Seine-Oise | 0.52 | 0.86 | -8.34 | 0.67 | 0.84 |
| Bourdet | Poitou-Charentes | 1.02 | 0.18 | -0.72 | -0.51 | 0.24 |

**Table 3: Classification of water table level classes related to the values of the SPLI corresponding to the MSH limits**

| Classification | SPLI values | Return periods |
|---|---|---|
| Very low groundwater level | < -1.28 | > 10 dry years |
| Low groundwater level | between -1.28 and -0.84 | Between 10 dry years and 5 dry years |





| Moderately low groundwater level | between -0.84 and -0.25 | Between 5 dry years and 2.5 dry years |
| Normal groundwater level | between -0.25 and 0.25 | Between 2.5 dry years and 2.5 wet years |
| Moderately high groundwater level | between 0.25 and 0.84 | Between 2.5 wet years and 5 wet years |
| High groundwater level | between 0.84 and 1.28 | Between 5 wet years and 10 wet years |
| Very high groundwater level | > 1.28 | > 10 wet years |

**Table 4: Statistical scores of the comparison between the simulated and observed daily evolution of river discharges shown in Figure 13.**

| Gauging Station | NSE | Correlation | Ratio |
|---|---|---|---|
| Charente at Jarnac | 0.45 | 0.72 | 0.81 |
| Somme at Abbeville | 0.69 | 0.86 | 0.92 |
| Seine at Poses | 0.86 | 0.93 | 1.01 |
| Loire at Montjean | 0.94 | 0.97 | 1.05 |