# Peer review of "The AquiFR hydrometeorological modelling platform as a tool for improving groundwater resource monitoring over France: evaluation over a 60-year period."

_Hydrology and Earth System Sciences, 2019_

## Referee Comment (RC1) · Anonymous Referee #1 · 3 Jul 2019

The manuscript presents a newly developed hydrometeorological modelling platform (AquiFR). AquiFR combines three groundwater flow models, one land surface model, and one meteorological model embedded in a coupling system. The modelling platform computes groundwater levels, river flow, and spring discharge. The presented work compares the output of the modelling platform with long-term measured time series from 11 aquifers and 22 karstic springs. Several quality criteria are applied in order to assess the model efficiency.

The implementation of a national-wide hydrometeorological modelling platform is an

important topic, especially for the assessment of future climate changes. The combination of multiple models having different model structures is an interesting approach. The manuscript gives a brief introduction of the incorporated models, but unfortunately, the model descriptions are not really detailed. This makes it difficult for the reader to understand how the newly developed modelling platform functions. However, the presentation of the result is sufficient for the aim of the manuscript. Overall, the model output is a good approximation of the measured time series. One negative aspect of the result section is the lack of structure. I recommend publication with minor revisions, including a thorough review to improve the language.

My detailed comments are listed below.

Main Comments

- The presentation of the AquiFR Hydrometeorological Modelling Platform is neither well described nor structured. Especially the first two paragraphs of Section 2, intended to be an introduction of the newly developed model, lacks a detailed description of the connection between different compartments. This part of the text should be closely connected to a meaningful (!) scheme of the AquiFR platform. I highly recommend replacing Figure 1 with a more detailed scheme and using this as a central theme guiding the reader through Section 2.

- "SURFEX is a modelling platform aiming to simulate the water and energy fluxes at the interface between the surface and the atmosphere" (Page 6, Line 5); "MARTHE embeds single to multilayer aquifers, hydrographic networks and the exchanges with the atmosphere (rainfall, snow and evapotranspiration) for the computation of the soil water balance" (Page 7, Line 24); "Snow accumulation, snow melting and pumping is taken into account" (EROS software; Page 7, Line 24) How do you deal with redundant parameters and processes which originally are elements in several of the models (e.g. evapotranspiration)?

- The SURFEX modelling platform: You are using the SURFEX model to calculate

groundwater recharge and surface runoff. How do you address the specific karstic features (e.g. Epikarst, fast recharge components) in your model?

- Why do you present the quality criteria in section 3 (Results)? I would like to have more information about the evaluation of the model quality: a) general descriptions of the applied criteria, b) information about the calculation (equations) and references: e.g. How do you define bias and how do you exclude the bias from the calculation of the normalized RMSE?

- The Numbering of the sections should be adapted. Section 4 is entirely missing.

Secondary Comments Introduction: The beginning of the Introduction has been kept general. I would like to have more information on "but is still poorly known" (Page 2, Line 2) and I do not understand what you mean by "Groundwater is indeed located at some depth below the soil" (Page 2, Line 2).

Page 2, Line 28: "[. . .] as separate layers discretized using a 5 km resolution grid [. . .]" – The word separate is confusing here. Please, rephrase this sentence and maybe the next one as well. Also point out that the different layers are not connected to each other but to the river network.

Page 3, Line 25: I do not understand how the AquiFR project can provide monitoring of groundwater resources. Please, elaborate this.

The SAFRAN meteorological reanalysis: I am not sure if "analyses eight variables" (Page 5, Line 26) and "analyses each atmospheric variables" (Page 5, Line 30) are suitable expressions. Although, Quintana-Seguí et al. (2008) uses the same expression. I think estimates or calculates would be more suitable here.

Page 6, Line 6: The sentence need to be rephrased.

Page 6, Line 9: "[. . .] SURFEX is used in offline mode [. . .]" If this part of the sentence is useful information for the reader, elaborate it. Otherwise, I would delete it.

Page 7, Line 18: "hydro-climatic rainfall-river flow-piezometric head distributed model" is the direct translation of the expression used in Thiéry (2018a). Don't you think "reservoir model" is also a correct description of the model?

Page 9, Line 22: Please, erase the brackets and use a different expression (e.g. wise versa) instead.

Page 12, Line 21: Please, consider rephrasing the sentence "They were kept [. . .]"

Page 13, Line 1: Please, consider splitting this sentence.

Page 13, Line 11: Why did you (re)calibrate a few of the catchment/karst models and others not? You are proposing an inverse calibration tool - How did you calibrate the models after connecting them to SURFEX?

Page 14, Line 18: What do you mean by "However, the SIM tool uses coarse hydrogeological modelling [. . .]"?

Figure 2/3: Karst springs or Karst instead of Karsts

MINOR COMMENTS AND TYPOGRAPHICAL ERRORS - Page 1, Line 27: to compute

- Page 1, Line 28: that is used

- Page 2, Line 1: on Earth

- Page 2, Line 15: research organizations (?)

- Page 2, Line 27: United Kingdom (UK)

- Page 3, Line 2: delete though

- Page 3, Line 13: on a global scale

- Page 3, Line 17: led by the

- Page 3, Line 18: delete Indeed

- Page 3, Line 25: the AquiFR project was initiated

- Page 3, Line 27: numerical modelling (?)

- Page 4, Line6: I am not sure if reported on the present study is a suitable expression: presented by?

- Page 4, Line 20: period. In

- Page 5, Line 26: eight variables: rainfall, snowfall

- Page 5, Line 26: air temperature and relative humidity 2 m (above ground) and wind speed 10 m above ground.

- Page 5, Line 29: two rain gauges. SAFRAN

- Page 6, Line 2: zone. Further

- Page 6, Line12: temporal

- Page 6, Line18: gathers numerical

- Page 6, Line 23: Horizontal groundwater flow (?)

- Page6, Line 24: leakage. Therefore

- Page6, Line 29: coupled to

- Page 7, Line 15: Thiéry et al, 2018 – a or b?

- Page 8, Line 25: Is there a number missing in the brackets?

- Page 9, Line 16: observations at

- Page 9, Line 26: 2m and 4 m, respectively.

- Page 10, Line 2: with at least

- Page 11, Line 19: model input instead of inputs of the model

- Page 11, Line 29: delete one of the two dots

- Page 12, Line 4: shows

- Page 12, Line 8: which refers to the extreme rainfall event at the end of May 2016.

- Page 12, Line 11: Better: Figure 12 shows two plots comparing. . .

- Page 12, Line 21: same here

- Page 13, Line 7: Nevertheless, some regions are

- Page 13, Line 9: than the other regions (cf. Fig. 5).

- Page 13, Line 25: It would also demand big resources of computational power.

- Page 13, Line 26: to simulate a

- Page 14, Line 8: into account the

- Page 15, Line 13: more regional models?

- Page 15, Line 18: in progress

- Figure 15: (b) Somme

---

## Referee Comment (RC2) · Anonymous Referee #2 · 22 Jul 2019

General comments ======================

This article presents a new platform to simulate groundwater in France. The platform puts together two different hydrogeological models and one lumped model system for karst aquifers. The objective is to be able to simulate the whole of France with one set of tools. At this stage, about half of France is implemented. This aims at being and extension of the SIM France hydrometeorological model. This is a very interesting approach to national-scale groundwater modeling and thus deserves publication. Furthermore, AquiFR will probably play an important role in water management and

research in France in the future, so the paper is needed.

The main strength of the AquiFR approach (a sort of confederation of models) is also its weaknesses (how do you generate homogeneity from such an heterogeneous set of models?). AquiFR is not one model, it is a set of models that simulate different aquifers using the same forcing. This has allowed to put a large scale high resolution hydrogeo-logical model together much faster than if a single model was run from scratch, which is very valuable. We should not forget that these models have a long history behind them and thus have had different feedback cycles with stakeholders. That would be very difficult to do from scratch. However, the very nature of AquiFR makes interpreting the results difficult, as different systems are simulated with different models and, thus, the results will not be spatially homogeneous. However, the model is validated almost as if it was one model, without discussing in detail the role of each of the models. In order to compensat for the heterogeneity, all the models are forced with the recharge that comes from SURFEX, which seems appropriate, but there are some exceptions, for example, in one case the partition between surface runoff and recharge is recalcu-lated with GARDENIA or in another the model is also fed by streamflow observations. Achieving homogeneity is difficult with so many different models and approaches. The problem is that, as the text is now, all this is a little bit confusing and it is very difficult to get a clear picture of how the model works.

As I wrote in the first paragraph, the article is valuable and deserves to be published, but first it must undergo an improvement of the text in order to clarify how the model works, how it has been calibrated and how it has been validated.

The article explains the model architecture, but not with enough detail. I have many questions on this topic that should be clarified in the text:

1. It is not clear if the runoff is calculated with a common code and/or grid (P15L17 suggest that it is not the case), but nothing is explained about how the river routing is performed in the different basins.

2. Are the rivers connected bidirectionally with the groundwater? P6L28-29 suggests it can be done, but is it done?

3. EROS are lumped models that simulate karst in a simpler way. This is reasonable. It is mentioned that AquiFR will be used for climate change studies, but it is not mentioned how the calibrations of these lumped models will hold in a changing climate.

4. It is not clear if there is a bidirectional coupling between the aquifer and the soil (SURFEX). P6L15 says it can be done. Figure 1 shows and arrow that goes from the post-processing to SAFRAN/SURFEX, but it is not clear what it means. Is there a bidirectional coupling between soil and aquifer? Is SURFEX just a forcing or at each time step it is updated with information coming from the aquifers?

5. It seems that all the models have been recalibrated in order to be able to use the recharge coming from SURFEX. However P13L9-11 confuses me on this point. Have the models been recalibrated in order to use SURFEX as forcing?

6. It is not discussed if the the recalibrated models forced by SURFEX perform better or worse than the same models, calibrated with P/ETP data and using P/ETP data as forcing. What is the impact of using SURFEX as forcing? Having a homogeneous forcing has value, but does it have downsides?

7. How good is the partition of surface runoff and drainage of SURFEX, in general? This is a key input for the whole system, but it is not validated, not even discussed in the paper. As far as I understand SURFEX may have some empirical parameters in order to determine surface runoff. Has this been calibrated? I would like to see a discussion (and data if possible) on the quality of the SURFEX recharge, as it is the main input for the hydrogeological models used in AquiFR.

8. In the Somme river you don't use SURFEX's partition between runoff and recharge. It seems, that GARDENIA (no citation is provided) adds them together and makes a new partition. Why? How? This should be explained.

[Figure]

9. Some applications of MARTHE need observed streamflow as an input (boundary conditions). How will you simulate climate change in this area? Why don't you use model streamflow? You simulate it, don't you? You should clarify this point.

I also have some questions on the cal/val procedure.

1. Have you calibrated all the models over the same time period? If no, why? Due to data availability?

2. Do you validate all the models over the whole 60 year period? Do you use the calibration data also for validation? Do you only validate on independent data? The text is not clear to me on this regard and this is a very important issue. Not only for heads, also for streamflow. A model should not be validated using the same data it was used for calibration. If this cannot be avoided, it must be well justified.

3. You show the metrics you used for validation, but not for calibration. I guess that each model is calibrated differently, using different tools. Is this the case? This should be commented.

4. You also validate using the NSE. Have you considered the KGE? Or even better, the non parametric version of the KGE (Pool et al, 2018)? The KGE allows to separate the contribution of the correlation, the bias and the standard deviation. The non parametric form makes less assumptions on the underlying data distribution so it can be used with different kinds of variables with less problems. Also, the non parametric form is less sensitive to extremes (so you would not need to calculated the sqrt of the streamflow, as you do). I guess it is too late to change this, but you should consider this in the future.

5. Could you explain with more detail what is the NRMSE-BE? Have you substracted the mean and divided by the standard deviation and then calculated the RMSE? Have you removed the seasonal signal? A little bit more detail on this unusual metric should be provided.

[Figure]

Using the SPLI is appropriate, but you don't detail enough how you calculate it.

1. Which method do you use to calculate the standardized series? Is it parametric or non parametric?

2. If it is parametric, which distribution do you fit your data to? Does it fit to all areas equally well?

3. Figure 10 shows the distribution of the different categories of the SPLI. But some of them are bimodal. I would expect a normal distribution as an standardized variable involves renormalizing the data to a normal distribution. Why these figures don't show a normal distribution?

I suggest adding a Methodology section where the cal/val procedure is presented and where the indicators (NRMSE-BE, NSE) and standardizations (SPLI) are presented.

Anthropic processes: You take pumping into account for some models. But the subsequent irrigation is not taken into account by SURFEX. Can you comment a little bit more on the current state of anthropic impacts in AquiFR and how this affects the results?

Specific comments ========================

* P2L6: "Thus, modeling is still a useful tool ...". Well, even with high resolution remote sensing data of storage in aquifers, models would still be useful, as they allow to connect aquifers with the rest of the system (soil, streams, etc.).

* PL1: "3 groundwater flow software" -> 3 groundwater flow models.

* P4L20: "period.In" -> period. In

* P6L17: "gathersnumerical" should be separated.

* P6L29: coupledto should be separated.

* P7L18: "set of rivers organized in sub-basins". Is this the basis of the acronym? I guess it is in its French form. Maybe it would be better to just put the French name.

[Figure]

* P8L14: GARDENIA (citation needed). You should also explain how GARDENIA works.

* P9L17: observationsat should be divided.

* P11L8: It sensitivity -> Its sensitivity.

* P12L18-20: So you validate on the same stations you used for calibration. Do you?

* P12L33-P13L1. You calculate the sqrt to avoid an excecive influence of extremes. Is this the case? You should explain it.

* P13L10-11: Here you imply that you didn't recalibrate the models in order to use SURFEX as forcing. But earlier it seems you did. Did you?

* P13L16-29: I would move this into the introduction.

* P14L6: Which periods were used for calibration?

* Fig1: What do the arrows mean? What fluxes are send to the post-processing and what is send back to SAFRAN/SURFEX? I would add labels to the arrows.

* Fig7: Put the legend outside of the first plot.

* Fig10: Being standardized values, I would expect a normal distribution, but on three cases it is bimodal.

* Fig11b: difficult to see the circles.

* Fig12: Why are the x-axis time scales so different? Is it related to data availability? Which is the calibration period?

---

## Author Comment (AC1) · 22 Aug 2019

Response to reviewer hess-2019-166 The AquiFR hydrometeorological modelling platform as a tool for improving groundwater resource monitoring over France: evaluation over a 60 year period.

**Anonymous Referee #1**

**Main Comments**

**Comments:** - The presentation of the AquiFR Hydrometeorological Modelling Platform is neither well described nor structured. Especially the first two paragraphs of Section 2, intended to be an introduction of the newly developed model, lacks a detailed description of the connection between different compartments. This part of the text should be closely connected to a meaningful (!) scheme of the AquiFR platform. I highly recommend replacing Figure 1 with a more detailed scheme and using this as a central theme guiding the reader through Section 2.

**Response:** Thanks for this comment. In order to improve this section, we added a paragraph that presents the physical connection between the compartments as well as a new scheme (see Figure 1 below). Moreover, we replaced the former Figure 1 by a more detailed scheme with a detailed description of the time step during an AquiFR simulation (see Figure 2 below). The description of the AquiFR platform is modified in consequence in section 2 in the revised manuscript:

"The AquiFR hydrometeorological modelling platform allows representing the main hydrological processes occurring within the watersheds from precipitations to groundwater flows as shown in **Erreur ! Source du renvoi introuvable.Erreur ! Source du renvoi introuvable.** AquiFR accounts for spatial heterogeneity by using different spatial scales. The atmospheric forcing from SAFRAN and the estimation of the surface water budget fluxes by SURFEX are provided on an 8 km resolution grid. The SAFRAN meteorological analysis (Quintana Segui et al., 2008) provides hourly precipitation (rainfall and snowfall), temperature, relative air humidity, wind speed and downward radiations. The SURFEX land surface model (Masson et al., 2013) needs these atmospheric variables to solve the energy and surface water budget at the land-atmosphere interface at a 5-minutes time step. SURFEX estimates the spatial partition of the flow between surface runoff and groundwater recharge. It accounts for different soil and vegetation types and uses a diffusion scheme to represent the transfer of heat and water through the soil. The soil in SURFEX is represented by a multilayer approach. Its depth varies according to vegetation (in France from 0.2 to 3m) and is partly accessible to plant roots. Deep soil infiltration constitutes groundwater recharge flux. Surface runoff can occur according to saturation excess.

The simulation of the watersheds depends on its hydrogeologic characteristics. For sedimentary basin, these two fluxes are transferred to the MARTHE (Thiéry, 2015) or EauDyssée (Saleh et al., 2013) groundwater models. These models simulate transfer to the unsaturated zone, groundwater flows within and between the aquifer layers, transfer of surface runoff to and within rivers, and riveraquifer exchanges. They also account for the numerous groundwater abstractions within the river basins. The temporal resolution is daily and the spatial resolution varies from 100 m to a maximum of 8000 m. The depth of the deepest aquifer layer can reach locally about 1000 meters.

Karstic aquifer systems are simulated through a conceptual reservoir modelling approach using the EROS software (Thiéry, 2018). Each karstic system is represented by a lumped-parameter reservoir model solved at a daily time scale. Conceptual approaches are preferred for simulating karstic systems since their heterogeneities make it difficult to use a physically-based approach. EROS uses the

[revised manuscript text omitted]

---

## Author Response (AR1)

Author's response for hess-2019-166
The AquiFR hydrometeorological modelling platform as a tool for improving groundwater resource monitoring over France: evaluation over a 60 year period.

**Anonymous Referee #1**

**Main Comments**

**Comments:** - *The presentation of the AquiFR Hydrometeorological Modelling Platform is neither well described nor structured. Especially the first two paragraphs of Section 2, intended to be an introduction of the newly developed model, lacks a detailed description of the connection between different compartments. This part of the text should be closely connected to a meaningful (!) scheme of the AquiFR platform. I highly recommend replacing Figure 1 with a more detailed scheme and using this as a central theme guiding the reader through Section 2.*

**Response:** Thanks for this comment. In order to improve this section, we added a paragraph that presents the physical connection between the compartments as well as a new scheme (see Figure 1 below). Moreover, we replaced the former Figure 1 by a more detailed scheme with a detailed description of the time step during an AquiFR simulation (see Figure 2 below). The description of the AquiFR platform is modified in consequence in section 2 in the revised manuscript:

[revised manuscript text omitted]

a) Computation of the groundwater recharge and surface runoff from SURFEX prior to an AquiFR run

b) Components of the OpenPALM application and workflow of an AquiFR run

*Figure 2: Scheme of the numerical implementation of AquiFR. (a) SAFRAN and SURFEX are run separately, as well as the processes that extract the daily surface runoff and groundwater recharge at 8 km resolution on a daily time step over the full 60 year period. (b) The components implemented within the coupling system O-Palm are presented. Pre-processing in blue gives access to the surface runoff and groundwater recharge as well as atmospheric forcing to the 3 groundwater models for the current time steps. Then, each hydrogeologic software runs all of their models for the current time step. The fluxes and state variables are then transferred daily to the post-processing, that writes the model outputs and manage the following time step.*

**Comment:** - *"SURFEX is a modelling platform aiming to simulate the water and energy fluxes at the interface between the surface and the atmosphere" (Page 6, Line 5); "MARTHE embeds single to multilayer aquifers, hydrographic networks and the exchanges with the atmosphere (rainfall, snow and evapotranspiration) for the computation of the soil water balance" (Page 7, Line 24); "Snow accumulation, snow melting and pumping is taken into account" (EROS software; Page 7, Line 24) How do you deal with redundant parameters and processes which originally are elements in several of the models (e.g. evapotranspiration)?*

**Response:** Indeed, some information was missing. The MARTHE hydrogeological software includes different options that can be used to generate surface runoff or groundwater recharge. It includes its own computation of the soil water balance, including evapotranspiration, surface runoff and recharge. It can also directly receive surface runoff and recharge from an independent model, that is SURFEX in our case. This is this second option that is used in AquiFR, and this is now stated explicitly in the text.

The EROS software is not connected to SURFEX, and is directly connected to SAFRAN, this is now more clearly explained in the new paragraph of section 2 and appears clearly in Figure 1.

**Comment:** - *The SURFEX modelling platform: You are using the SURFEX model to calculate groundwater recharge and surface runoff. How do you address the specific karstic features (e.g. Epikarst, fast recharge components) in your model?*

**Response:** Specific karstic features are not taken into account in the SURFEX land surface model. Epikarst and fast recharge components could affect the simulation of karstic flows in SURFEX.  This is why EROS is not connected to SURFEX and instead uses directly the atmospheric forcing from SAFRAN. The new paragraph in section 2 and the additional scheme better present the multilayer aspect of SURFEX and the main characteristics of the way the runoff and infiltration are computed. A few words about this are now added to the manuscript (section 2.2):

*"The soil column thickness represented in each 8 km resolution grid cell varies from 20 centimeters to 3 meters according to the land cover in France and mostly corresponds to the root zone layer (Decharme et al., 2013). Thus, the recharge provided by SURFEX is the vertical flux leaving the bottom of the soil column of each grid cell. Further details on ISBA can be found in Decharme et al. (2013)."*

**Comment:** - *Why do you present the quality criteria in section 3 (Results)? I would like to have more information about the evaluation of the model quality: a) general descriptions of the applied criteria, b) information about the calculation (equations) and references: e.g. How do you define bias and how do you exclude the bias from the calculation of the normalized RMSE?*

**Response:** In order to clarify the quality criteria used in section 3, a new Methodology section is now included in the revised manuscript. This methodology section includes 3 subsections:

3.1 The regional models implemented in the AquiFR platform

3.2 Calibration of the hydrogeological models

3.3 Evaluation criteria of the 60 years long-term simulation

 This last subsection includes a general description of the applied criteria, that is bias, Nash-Sutcliff coefficient, normalized RMSE bias-excluded, and SPLI indicator. This new section is presented at the end of the present document.

**Comment:** - *The Numbering of the sections should be adapted. Section 4 is entirely missing*.

**Response:** It is now corrected in the revised manuscript.

**Secondary Comments**

**Comment:** *Introduction: The beginning of the Introduction has been kept general. I would like to have more information on "but is still poorly known" (Page 2, Line 2) and I do not understand what you mean by "Groundwater is indeed located at some depth below the soil" (Page 2, Line 2).*

**Response:** We agree about this. The beginning of the introduction was changed in order to be more explicit about the context:

*"Groundwater is the most important freshwater resources on Earth. It is widely used for drinking water, agricultural, and industrial use. Knowing the spatial and temporal evolutions of the groundwater and being able to predict its future evolution over short to long term periods are essential to manage water resources and anticipate climate change impacts. However, groundwater is characterized by a strong spatial heterogeneity making its monitoring difficult. Thus, it is mostly monitored through well networks that can give information only at specific locations (Aeschbach-Hertig and Gleeson, 2012; Fan et al., 2013). Remote sensing gravimeters can provide large scale estimates of groundwater storage changes (Long et al., 2015) but it is not suited for regional scale studies (Longuevergne et al., 2010). Therefore, modeling can be a useful tool to provide meaningful information on the groundwater resources (Aeschbach-Hertig and Gleeson, 2012) at different spatial scales and different temporal periods in the past or in the future."*

**Comment:** *Page 2, Line 28: "[: : :] as separate layers discretized using a 5 km resolution grid [: : :]" – The word separate is confusing here. Please, rephrase this sentence and maybe the next one as well. Also point out that the different layers are not connected to each other but to the river network.*

**Response:** The authors agree that this part was not clear. It is modified in the revised manuscript as follows: *"In the United Kingdom (UK), Pachocka et al. (2015) used a numerical model to compute the piezometric head evolution of the three most important UK unconfined aquifers using a finite difference scheme. These three unconfined aquifer basins were discretized into a 5 km resolution grid and connected to a river network. The model was tested against 37 gauging stations distributed across the country."*

**Comment:** *Page 3, Line 25: I do not understand how the AquiFR project can provide monitoring of groundwater resources. Please, elaborate this.*

**Response:** AquiFR is expecting to help monitoring the groundwater resources since it is planned to be used on real-time, in order to provide each day a present state of groundwater on the simulated domain. The sentence was modified as follows:" *In such context, the AquiFR project was initiated to capitalize these developments in order to provide real time monitoring (Coustau et al., 2015); and forecasts (Singla et al., 2012; Thirel et al., 2010) of groundwater resources in France, as well as long-term reanalyses and future projections"*

**Comment:** *The SAFRAN meteorological reanalysis: I am not sure if "analyses eight variables" (Page 5, Line 26) and "analyses each atmospheric variables" (Page 5, Line 30) are suitable expressions. Although, Quintana-Seguí et al. (2008) uses the same expression. I think estimates or calculates would be more suitable here.*

**Response:** The authors agree. We changed "analyses" by "estimates".

**Comment:** *Page 6, Line 6: The sentence needs to be rephrased.*

**Response:** The sentence is now: "*SURFEX is built to be coupled to forecast and climate models. It includes databases and interpolation scheme and several physical options that allows to use it at different spatial and temporal scales*"

**Comment:** *Page 6, Line 9: "[: : :] SURFEX is used in offline mode [: : :]" If this part of the sentence is useful information for the reader, elaborate it. Otherwise, I would delete it.*

**Response:** This part was modified. The new sentence gives more information on the coupling between SURFEX and the aquifers. The part "offline mode" is deleted: "*In the present study, no bidirectional coupling between the soil of SURFEX and the aquifers is taken into account. Thus, a one-way coupling from the soil of SURFEX to the aquifer is taken into account in order to provide*
5      *groundwater recharge and surface runoff to the AquiFR platform*"

**Comment**: *Page 7, Line 18: "hydro-climatic rainfall-river flow-piezometric head distributed model" is the direct translation of the expression used in Thiéry (2018a). Don't you think "reservoir model" is also a correct description of the model?*

**Response:** We agree with the reviewer. We replaced this expression by "distributed reservoir model"

10   **Comment:** *Page 9, Line 22: Please, erase the brackets and use a different expression (e.g. wise versa) instead.*

**Response:** The text is now "*A positive value means that the simulation overestimates the mean piezometric head with respect to the observation while a negative value means the opposite.*"

**Comment:** *Page 12, Line 21: Please, consider rephrasing the sentence "They were kept [: : :]"*

15   **Response:** The new sentence is now: "*The present study used the same observed datasets to evaluate the river discharges simulated with the AquiFR platform  over the 1958-2018 period*"

**Comment:** *Page 13, Line 1: Please, consider splitting this sentence.*

**Response:** This sentence is now split: "*Regarding the results of Figure 15c, for rivers in continuous aquifers, 27% of the NSE scores are greater than 0.7. Moreover 58% of these NSE scores are greater*
20   *than 0.5 while 22% are negatives.*"

**Comment:** *Page 14, Line 11: Why did you (re)calibrate a few of the catchment/karst models and others not? You are proposing an inverse calibration tool - How did you calibrate the models after connecting them to SURFEX?*

**Response:** All the karst models were calibrated using the SAFRAN atmospheric forcing. Almost all the
25   distributed models included in AquiFR were calibrated using the SAFRAN-SURFEX fluxes. Some models were  not recalibrated either because the results were good enough with the new fluxes, or because additional changes are expected. Each distributed model was developed independently and calibrated with different periods of calibration. The calibration was based on  trial-and-error method over the same period that was used to develop them. To better address such question, a subsection
30   on the calibration is now presented in the new section "3. Methodology" at the end of the present document, and the Table 1 provides information on the  calibration.

**Comment:** *Page 14, Line 18: What do you mean by "However, the SIM tool uses coarse hydrogeological modelling [: : :]"?*

**Response:** In SIM, only few aquifers are simulated explicitly with the MODCOU hydrogeological
35   model: the Seine and the Rhône aquifer basins (Habets et al., 2008). These two models correspond to outdated versions that have not been upgraded since. Thus, in SIM, the Seine aquifers are described by only 3 aquifer layers rather in AquiFR, 6 layers are accounted for as well as the river loss to the aquifer. More details regarding this point is now added in the article: "*However, the SIM tool uses coarse hydrogeological modelling with less aquifer layers or no river loss to the aquifer. It mainly*
40   *focuses on operational forecasts of river flows and soil humidity.*"

**Comment:** *Figure 2/3: Karst springs or Karst instead of Karsts*

**Response:** It is corrected.

MINOR COMMENTS AND TYPOGRAPHICAL ERRORS
**Comment:** - *Page 1, Line 27: to compute*

**Comment:** - *Page 1, Line 28: that is used*

**Comment:** - *Page 2, Line 1: on Earth*

**Comment:** - *Page 2: Line 15: research organizations (?)*

**Comment:** - *Page 2, Line 27: United Kingdom (UK)*

**Comment:** - *Page 3, Line 2: delete though*

**Comment:** - *Page 3, Line 13: on a global scale*

**Comment:** - *Page 3, Line 17: led by the*

**Comment:** - *Page 3, Line 18: delete Indeed*

**Comment:** - *Page 3, Line 25: the AquiFR project was initiated*

**Comment:** - *Page 3, Line 27: numerical modelling (?)*

**Comment:** - *Page 4, Line6: I am not sure if reported on the present study is a suitable expression: presented by?*

**Comment:** - *Page 4, Line 20: period. In*

**Comment:** - *Page 5, Line 26: eight variables: rainfall, snowfall*

**Comment:** - *Page 5, Line 26: air temperature and relative humidity 2 m (above ground) and wind speed 10 m above ground.*

**Comment:** - *Page 5, Line 29: two rain gauges. SAFRAN*

**Comment:** - *Page 6, Line 2: zone. Further*

**Comment:** - *Page 6, Line12: temporal*

**Comment:** - *Page 6, Line18: gathers numerical*

**Comment:** - *Page 6, Line 23: Horizontal groundwater flow (?)*

**Comment:** - *Page6, Line 24: leakage. Therefore*

**Comment:** - *Page6, Line 29: coupled to*

**Response:** Thanks for all these corrections. They are now corrected in the revised manuscript.

**Comment:** - *Page 7, Line 15: Thiéry et al, 2018 – a or b?*

**Response:** Thiéry et al., 2018  is the correct citation ; it corresponds to the reference Thiéry, D., Amraoui, N. and Noyer, M.-L.: Modelling flow and heat transfer through unsaturated chalk – Validation with experimental data from the ground surface to the aquifer, J. Hydrol., 556, 660–673, doi:10.1016/j.jhydrol.2017.11.041, 2018

The citations Thiéry 2018a, and Thiéry 2018b correspond to the references with only Thiéry in single author:

Thiéry, D.: Logiciel ÉROS version 7.1 - Guide d'utilisation. Rapport final, BRGM/RP-67704-FR, Orléans., 2018a.

5 Thiéry, D.: Modélisation hydrologique globale des débits de 23 sources karstiques avec le logiciel ÉROS. Rapport final, BRGM/RP-67723-FR, Orléans., 2018b.

**Comment:** - *Page 8, Line 25: Is there a number missing in the brackets?*

**Response:** yes, we wanted to gives the estimation of 16 mm/year in billion of m3 per year (that is 2.4 billion of m3 per year). Thank you for this correction.

10 **Comment:** - *Page 9, Line 16: observations at*

**Comment:** - *Page 9, Line 26: 2m and 4 m, respectively.*

**Comment:** - *Page 10, Line 2: with at least*

**Comment:** - *Page 11, Line 19: model input instead of inputs of the model*

**Comment:** - *Page 11, Line 29: delete one of the two dots*

15 **Comment:** - *Page 12, Line 4: shows*

**Comment:** - *Page 12, Line 8: which refers to the extreme rainfall event at the end of May 2016.*

**Comment:** - *Page 12, Line 11: Better: Figure 12 shows two plots comparing*

**Comment:** - *Page 12, Line 21: same here*

**Comment:** - *Page 13, Line 7: Nevertheless, some regions are*

20 **Comment:** - *Page 13, Line 9: than the other regions (cf. Fig. 5).*

**Comment:** - *Page 13, Line 25: It would also demand big resources of computational power.*

**Comment:** - *Page 13, Line 26: to simulate a*

**Response:** All these elements are corrected in the revised manuscript.

**Comment:** - *Page 14, Line 8: into account the*

25 **Response:** "into account the" instead of "into account in the"

**Comment:** - *Page 15, Line 13: more regional models?*

**Response**: "more regional model" instead of "more regional spatial model"

**Comment:** - *Page 15, Line 18: in progress*

**Comment:** - *Figure 15: (b) Somme*

30 **Response:** All these elements are corrected in the revised manuscript.

**Anonymous Referee #2**

**General comments**

**Comment**: *1 It is not clear if the runoff is calculated with a common code and/or grid (P15L17 suggest that it is not the case), but nothing is explained about how the river routing is performed in the different basins.*

**Response:** Yes, indeed, some information was lacking.  Surface runoff is computed using the land surface scheme of SURFEX on an 8 km resolution grid. This 8 km resolution grid corresponds to the grid provided by the SAFRAN atmospheric analysis. The surface runoff is then routed to the river by the hydrogeological models, with their own spatial resolution (varying from 100 m to 8 km). To be clearer, in the revised manuscript, the section 2 is entirely modified. It includes a new scheme (see Figure 1 of the present document) presenting the physical interaction between the modules and the main processes accounted for the estimation of the water flows, and on a new version of the former Figure 1 (see Figure 2 of the present document) that better presents the technical connection between the module. The computation of river routing is described in section 2.4 for Marthe and 2.3 for EauDyssée.

**Comment:** *2. Are the rivers connected bidirectionally with the groundwater? P6L28-29 suggests it can be done, but is it done?*

**Response:** Yes rivers are connected in both direction in the MARTHE and EauDyssée models. A sentence is added in the text (see answer above). This information is also now provided in section 2.3 (EauDyssée) and 2.4 (MARTHE) that presents the hydrogeological models and is shown in the new Figure 1.

**Comment:** *3. EROS are lumped models that simulate karst in a simpler way. This is reasonable. It is mentioned that AquiFR will be used for climate change studies, but it is not mentioned how the calibrations of these lumped models will hold in a changing climate.*

**Response**: A new section "3. Methodology" is added to the manuscript for describing the models, the calibrations and the statistical criterias used for the evaluation. A subsection of this new section "3. Methodology" is now devoted to the calibration of the models. It is now stated that "*For the karst system software EROS, the models were calibrated based on the SAFRAN atmospheric analysis, by using an optimization of the statistical comparison between observed and simulated daily riverflows.*" This new section is presented at the end of this document.

It is true that part of the uncertainty of the impact of climate change on the karst systems is linked to the hydrological model and to its calibration.  But it is beyond the scope of this article to discuss such uncertainty.

**Comment:** *4. It is not clear if there is a bidirectional coupling between the aquifer and the soil (SURFEX). P6L15 says it can be done. Figure 1 shows and arrow that goes from the post-processing to SAFRAN/SURFEX, but it is not clear what it means. Is there a bidirectional coupling between soil and aquifer? Is SURFEX just a forcing or at each time step it is updated with information coming from the aquifers?*

**Response:** Thanks to stress out this important point. Although capillary rise can be accounted for in SURFEX, in the current version of AquiFR, no bidirectional coupling between the aquifer and the soil is taken into account. This is no clearly stated in section 2: "*In this version 1.2 of AquiFR, no feedback from groundwater to the soil of SURFEX is taken into account. Therefore, a preliminary step illustrated by Figure 2a  is to estimate groundwater recharge and surface runoff with SURFEX taking*

*into account the atmospheric forcing from SAFRAN prior to an OpenPALM run".* In section 2.2 presenting SURFEX, it is now stated: "*In the present study, no bidirectional coupling between the soil of SURFEX and the aquifers is taken into account. Thus, a one-way coupling from the soil of SURFEX to the aquifer is taken into account in order to provide groundwater recharge and surface runoff to the AquiFR platform.* ". The former Figure 1 was modified to better explain what are the exchanged data between each modules within the AquiFR platform.

**Comment:** *5. It seems that all the models have been recalibrated in order to be able to use the recharge coming from SURFEX. However P13L9-11 confuses me on this point. Have the models been recalibrated in order to use SURFEX as forcing?*

**Response:** Yes, some information on the need of such calibration was stated page 4 lines 14-17, and is now even made clearer: "*the combined use of SURFEX and SAFRAN provides a consistent set of hydro-meteorological data over an 8 km resolution grid over France, including groundwater recharge and surface runoff from SURFEX, as well as potential evapotranspiration, precipitation, and temperature from SAFRAN. The use of these SURFEX 8 km resolution fluxes made necessary the recalibration of the hydrogeological models included in the platform*". Indeed, it was found that most often, there are some differences between the fluxes estimated by SURFEX and by the original water balance scheme using P/PET, mostly in terms of dynamic. Such differences affected some comparisons between observed and simulated heads, either positively or negatively. To give more information on this recalibration, a new subsection is now added in the new "Methodology" section 3 presented at the end of this document.

**Comment:** *6. It is not discussed if the recalibrated models forced by SURFEX perform better or worse than the same models, calibrated with P/ETP data and using P/ETP data as forcing. What is the impact of using SURFEX as forcing? Having a homogeneous forcing has value, but does it have downsides?*

**Response:** This is a good question, and it is true that no information was given in the first version of the article. Overall, the statistical results obtained with SURFEX were similar to those obtain with the original version. A sentence is now added to stress out this point (see answer above), and some information is added in Table 1. Such result is indeed disappointing, as SURFEX is a more physical model and is more demanding computationally. It is one objective of the AquiFR project to improve such results.

**Comment:** *7. How good is the partition of surface runoff and drainage of SURFEX, in general? This is a key input for the whole system, but it is not validated, not even discussed in the paper. As far as I understand SURFEX may have some empirical parameters in order to determine surface runoff. Has this been calibrated? I would like to see a discussion (and data if possible) on the quality of the SURFEX recharge, as it is the main input for the hydrogeological models used in AquiFR.*

**Response:** The SURFEX partition of the surface runoff and drainage may differ from those calculated by the original models. However, it is difficult to distinguish which of the two is closest to the truth, since the truth is unknown, and as, after recalibration, the statistical results obtained by the two versions are similar. It was necessary to modify the partition between surface runoff and drainage only for the Somme basin by using the total runoff. Comments on this point are now added in section 3 (provided at the end of the text) as well as in Table 1. Detailed information is provided in a report accessible online (Habets et al., 2017).

**Comment:** *8. In the Somme river you don't use SURFEX's partition between runoff and recharge. It seems, that GARDENIA (no citation is provided) adds them together and makes a new partition. Why? How? This should be explained.*

**Response:** It is now clearly stated that the partition between surface runoff and groundwater recharge in the Somme basin was biased by SURFEX with an overestimation of surface runoff in the North and an underestimation in the South. GARDENIA is the name of the water balance scheme used originally in MARTHE. But, to avoid confusion, we removed the name, added a reference, and some explanations on how it works: " *In order to compensate for this imbalance the total runoff provided by SURFEX was split into surface runoff and groundwater recharge using the original water balance scheme of MARTHE. This water balance scheme is based on a reservoir approach (Thiéry, 2014), for which the parameters were calibrated. Only one reservoir was used, enabling to modify the partition of the surface runoff, and to account for a delay on the groundwater recharge in order to mimic the impact of the deep unsaturated zone.*" In details, the reservoir we used is depicted below. H is the head in the reservoir, and is filled by the total runoff from SURFEX. THG is a time transfer coefficient and RUIPER is a partition coefficient that was calibrated. Using such reservoir, not only the partition of the flow between surface runoff and groundwater recharge is modified, but also the dynamic of the flow. This is important in the Somme basin since there is a deep unsaturated zone that is not simulated explicitly in the MARTHE model (see for instance Habets et al., 2010, Multi-model comparison of a major flood in the groundwater-fed basin of the Somme River (France), HESS)

[Figure]

$$ALIMG = H.dt/THG$$

$$QH = H.dt/(THG.RUIPER/H)$$

Figure 3 Partition of the total runoff of Surfex in the MARTHE Somme basin

The figure below presents the comparison of the river flow observed and simulated with Surfex with and without a new partition of the total runoff.

[Figure]

Figure 4 Comparison of the river flows at the outlet of the Somme basin between observations (blue) raw simulation with SAFRAN-SURFEX (orange) and simulation with the total runoff estimated by SAFRAN SURFEX and a partition of the surface runoff and drainage based on the MARTHE original water balance scheme

**Comment:** *9. Some applications of MARTHE need observed streamflow as an input (boundary conditions). How will you simulate climate change in this area? Why don't you use model streamflow? You simulate it, don't you? You should clarify this point.*

**Response:** In MARTHE, model streamflow are not simulated outside the simulated aquifer domain. Therefore, if the model does not encompass the entire river basin, boundary conditions are needed to impose flow on these rivers. We used observed streamflow in this version of AquiFR, but it is planned to use a new modelling method based on a lumped-parameter rainfall-runoff model to provide upstream river flows. The text is now modified: *"In the near future, the advantage to have the atmospheric forcing and surface fluxes over the entire domain will be used to estimate the upstream flow based either on a lumped-parameter rainfall-runoff model integrated in the MARTHE computer code or by the RAPID river routing model using a fine scale river network covering all France."*

Indeed, we have a hydrographic network over France, that is used for instance in SIM, but it has a 1km resolution, which is often not enough to match with rivers that are not fully simulated in the hydrogeological models.

In the climate change simulation we have done yet, the hypothesis is to have stable boundary conditions. Therefore, the flow of these not-fully simulated rivers, but also, the sea level, and the surface and groundwater abstractions are expected to be the same as in present day. Of course, it is clear that these hypotheses are not valid, and that the results only provide a first order impact of climate change.

*I also have some questions on the cal/val procedure.*

**Comment:** *1. Have you calibrated all the models over the same time period? If no, why? Due to data availability?*

**Response:** That is correct. The choice was made to calibrate on the same period used by the original model. This ensures that all the data needed are available, and allows comparing fairly with the original models. Please, report to the new section 3.2 provided at the end of the document.

**Comment:** *2. Do you validate all the models over the whole 60 year period? Do you use the calibration data also for validation? Do you only validate on independent data? The text is not clear to me on this regard and this is a very important issue. Not only for heads, also for streamflow. A model should not be validated using the same data it was used for calibration. If this cannot be avoided, it must be well justified.*

**Response:** The models are evaluated over the whole 60 year period. As described in the new Methodology section, the calibration procedure was done for each model using the same calibration period that were used to develop each model (see references in Table 1 and (Habets et al., 2017)). The new methodology section helps to better explain this. However, the validation presented in the article covers the 60-year period, restricted to the availability of the observation. Thus, the calibration and validation periods are different, but the validation period encompasses the calibration period. As all the models were not calibrated on the same period, and as the temporal availability of each measurement varies, it was the only way to have a full assessment of the whole AquiFR platform.

**Comment:** *3. You show the metrics you used for validation, but not for calibration. I guess that each model is calibrated differently, using different tools. Is this the case? This should be commented.*

**Response:** All the models were calibrated using the same statistical criteria: Efficiency, correlation and ratio for stream flow, and RMSE and biases for piezometric heads. As stated in the article, no automatic calibration tools were used, but only the skill of hydrogeological experts. These two points are now more clearly stated in the new 3. Methodology section : *"Hydrodynamic parameters, including hydraulic conductivities and specific yields, were modified based on hydrogeological expertise in order to obtain the best fit between observations and simulations. The calibration was made only on the piezometric heads, except for the MARTHE Somme model for which piezometric heads and riverflows were accounted for, and for the kartsic systems with karst spring flows only. All the models were recalibrated using the same statitiscal criterias."*

**Comment:** *4. You also validate using the NSE. Have you considered the KGE? Or even better, the non parametric version of the KGE (Pool et al, 2018)? The KGE allows to separate the contribution of the correlation, the bias and the standard deviation. The non parametric form makes less assumptions on the underlying data distribution so it can be used with different kinds of variables with less problems. Also, the non parametric form is less sensitive to extremes (so you would not need to calculated the sqrt of the streamflow, as you do). I guess it is too late to change this, but you should consider this in the future.*

**Response:** Thank you for this comment. Indeed, it is true that the KGE has some advantages compared to the NSE. This is a point that we will consider in the future. A sentence is added in the discussion : *"Some statistical scores using less assumptions on the underlying data distribution, such as the non-parametric variant of the Klunge-Gupta efficiency score, could be used to reduce the sensitivity to the extremes (Pool et al., 2018)."*

**Comment:** *5. Could you explain with more detail what is the NRMSE-BE? Have you substracted the mean and divided by the standard deviation and then calculated the RMSE? Have you removed the seasonal signal? A little bit more detail on this unusual metric should be provided*

**Response:** The details are now given in the new subsection 3.2 (see the methodology section at the end of this document).

**Comment:** *1. Which method do you use to calculate the standardized series? Is it parametric or non parametric?*

**Response:** The calculation of the Standardized Piezometric Level Index is similar to the calculation of the Standardized Precipitation Index (SPI) (McKee et al., 1993). The SPLI is an indicator used in the Monthly Hydrological Survey published each month. Details about its computation are given in Seguin, (2015). Considering a piezometric head time series of N years, the steps are the following:

- Step 1 : the monthly mean observed time series is computed
- Step 2 : constitution of twelve monthly time series (January to December) over the N year period. For each time series of N values, a non-parametric kernel density estimation (KDE) allows estimating the best probability density function (pdf) fitting the observed histograms. The SPI uses a gamma distribution, but time series of piezometric heads show a big variety of histogram. Therefore, the use of a KDE to estimate a pdf fitting the observed histogram is preferred.
- Step 3 : For each month from January to December, the adjusted pdf is projected over the standardized normal distribution using a quantile-quantile projection.

Figure 3 of the present document shows the procedure for the Omiécourt piezometer. The KDE helps to obtain a fit of the probability density function from the observed histogram. The cumulative

density function is deduced, and a projection over the standardized normal distribution allows deducing the SPLI.

[Figure]

Figure 3: Computation of the SPLI for the Omiécourt piezometer. The probability density function is estimated using kernel density estimator from observed monthly piezometric head values. The estimated cumulative density function is then estimated from the fitted pdf, and a quantile-quantile projection on the standardized normal distribution allows computing the SPLI.

The authors agree that the presentation of the SPLI was not detailed enough. A new presentation is proposed in the new section 3.3 Methodology.

**Comment:** *2. If it is parametric, which distribution do you fit your data to? Does it fit to all areas equally well?*

**Response:** As the observed distribution depends on the area where the piezometer is located, a non-parametric KDE is used to estimate the best pdf to fit the observations (Seguin, 2015).

**Comment:** *3. Figure 10 shows the distribution of the different categories of the SPLI. But some of them are bimodal. I would expect a normal distribution as a standardized variable involves renormalizing the data to a normal distribution. Why these figures don't show a normal distribution?*

**Response:** Figure 3 of the present document shows the histogram of the observed values which is bimodal. The fitted pdf from KDE is also bimodal, and therefore its projection on the standardized normal distribution will keep this bimodal characteristic.

**Comment:** *I suggest adding a Methodology section where the cal/val procedure is presented and where the indicators (NRMSE-BE, NSE) and standardizations (SPLI) are presented.*

**Response:** A new methodology section is added to the revised manuscript, including a presentation of these indicators.

**Comment:** *Anthropic processes: You take pumping into account for some models. But the subsequent irrigation is not taken into account by SURFEX. Can you comment a little bit more on the current state of anthropic impacts in AquiFR and how this affects the results?*

**Response:** Most of the groundwater abstraction is used for drinking water. Crop irrigation is not taken into account in the present version of AquiFR. This process can be activated in SURFEX. However, it involves setting up strong hypotheses (where are the irrigated fields, what are the irrigated volume for each field, and when is the irrigation provided) that are beyond the scope of the purpose of the evaluation proposed in this paper. As for the bidirectional coupling between groundwater and SURFEX, this is an option that could be used in the future development of AquiFR.

**Specific comments**

* P2L6: "Thus, modeling is still a useful tool ...". Well, even with high resolution remote sensing data of storage in aquifers, models would still be useful, as they allow to connect aquifers with the rest of the system (soil, streams, etc.).

**Response:** The author agree. This sentence is changed into *"At these regional scales, modeling can be a useful tool to provide meaningful information on the groundwater resources (Aeschbach-Hertig and Gleeson, 2012)."*

* PL1: "3 groundwater flow software" -> 3 groundwater flow models.

**Response:** We try to distinguish software (numerical code) from models (regional models ). For example, MARTHE is a hydrogeological modeling software, and 5 models have been developed using this software: the Somme, Poitou-Charentes, Nord-Pas-de-Calais, Basse-Normandie and Alsace models.

"3 groundwater flow software" is replaced by "3 hydrogeological modelling software"

* P4L20: "period.In" -> period. In

* P6L17: "gathersnumerical" should be separated.

* P6L29: coupledto should be separated.

**Response:** All these remarks are now corrected.

* P7L18: "set of rivers organized in sub-basins". Is this the basis of the acronym? I guess it is in its French form. Maybe it would be better to just put the French name.

**Response:** It is the english translation of the French acronym. We kept the french name, and as a consequence, we do the same for all the other acronyms that is SAM and MARTHE.

\* P8L14: GARDENIA (citation needed). You should also explain how GARDENIA works.

**Response:** We decided not to provide the name GARDENIA and only to keep the reference to the simplified water balance scheme that is implemented in MARTHE. It is effectively true that this water balance scheme is the same in the GARDENIA software, but it is fully implemented in the MARTHE software and it is now part of MARTHE. The new sentence is: *"This water balance scheme is based on a reservoir for which parameters are calibrated in order to compute the main components of the surface water budget (Thiéry, 2014)."*

\* P9L17: observationsat should be divided.

\* P11L8: It sensitivity -> Its sensitivity.

**Response:** All these remarks are corrected.

\* P12L18-20: So you validate on the same stations you used for calibration. Do you?

**Response:** Yes we do.

\* P12L33-P13L1. You calculate the sqrt to avoid an excecive influence of extremes. Is this the case? You should explain it.

**Response:** Yes. It is explained P12L17 to P12L20. We add a brief reminder about this.

\* P13L10-11: Here you imply that you didn't recalibrate the models in order to use SURFEX as forcing. But earlier it seems you did. Did you?

**Response:** Yes we did. See previous answer and section 3.2

\* P13L16-29: I would move this into the introduction.

**Response:** We decided to keep this part in the discussion in order to better highlight the choice of gathering several models in AquiFR as previously shown in section 2.

\* P14L6: Which periods were used for calibration?

**Response:** Periods for calibration were those initially used for calibrated each model independently. This is now better explained in the new Methodology section included in the revised manuscript. This particular part of the discussion

\* Fig1: What do the arrows mean? What fluxes are send to the post-processing and what is send back to SAFRAN/SURFEX? I would add labels to the arrows.

**Response:** The arrows illustrated the flux exchanges. The new proposed scheme better explains this.

\* Fig7: Put the legend outside of the first plot.

**Response**: The legend is now outside of the first plot.

\* Fig10: Being standardized values, I would expect a normal distribution, but on three cases it is bimodal.

**Response:** A new explanation of the SPLI indicators in the new methodology section helps to understand this.

* Fig11b: difficult to see the circles.

**Response:** The background SPLI map is now more transparent in order to better highlight the circles.

* Fig12: Why are the x-axis time scales so different? Is it related to data availability? Which is the calibration period?

5    **Response:** x-axis time scales are different because the axis limits are related to the observed data availability which is different for each karstic system. The calibration period corresponds to these axis limits. In order to be consistent with the evaluation of AquiFR, all the 60-year time serie is now shown.

**New section 3  Methodology**

[revised manuscript text omitted]

**Relevante changes made in the manuscript**

This part lists all the relevant changes made in the manuscript. It does not include corrections of typos as well as reformulations of sentences to improve the language that were included in the revised manuscript. The page, line and figure numbers correspond to the new revised manuscript:

P1L5 : " Longuevergne" was corrected in "Longuevergnes"

P1L11: " Ecole Normale Supérieure" was corrected in "Ecole normale supérieure".

P5L1 – P6L14: This paragraph of section 2 was modified for improving the description of the AquiFR platform. Former Figure 1 was replaced by two new figures: Figure 1 for the description of the physical processes taking place in AquiFR and Figure 2 showing the workflow of an AquiFR run.

P9L1 – P12L6: A new methodology section is added to present the regional models included in the AquiFR platform, the calibration of the models and the evalutation criteria used in the study.

P12L20: The number of gauging stations used to evaluate the simulated river flows of AquiFR was increased from 228 in the initial manuscript to 362. This increase concerns the EauDyssée Seine model as shown in the new Figure 15. With these new observations, the spatial distribution of river flow observations is more representative. The new observations were selected from the HYDRO database (http://hydro.eaufrance.fr/).

P12L27:  After a carefull check of the results, a few percentages presented in the results section of the initial manuscript were adjusted in the revised manuscript. They are listed below and concern minor changes that do not affect the overall results. Here, the value of 40% of the biases under 2 m was modified to 42%.

P12L29: 61% was replaced by 62%

P13L7 – L8: A mistake was detected in the computation of the criteria of the Omiécourt piezometer. The new RMSE_BE value is 0.93 instead of 0.81 while the bias is equal to -0.86 m instead of -0.15 m. The comments were changed accordingly and the changes are reported in Table 2.

P13L25 – L26: Percentages for the spatial distribution of the SPLI criteria were slightly adjusted. The new sentence is: "*20% of the NSE are greater than 0.7 and 56% greater than 0.5, while 12% are lower than zero. In parallel, 65% of the correlation are greater than 0.7."*

The old sentence was: *"20% of the NSE scores are greater than 0.7 and 55% greater than 0.5, while 11% are lower than zero. In parallel, 64% of the correlation scores are greater than 0.7"*

P14L13: After a carefull check of the results, percentages of categorized SPLI for the simulated piezometers were corrected with respect to the original manuscript. The new sentence is: *"19% (29%) of the simulated (observed) piezometers are in normal conditions, 46% (31%) are in moderately wet conditions, 16% (13%) are in wet conditions, and 16% (18%) are in extremely wet conditions."*

The old sentence was: *"17% (29%) of the simulated (observed) piezometers are in normal conditions, 50% (32%) are in moderately wet conditions, 14% (12%) are in wet conditions, and 16% (18%) are in extremely wet conditions"*

P15L11: In order to be more representative of the results, the number of gauging stations for evaluating river flows was increased from 228 to 362. New gauging stations concern the Seine model. New results for NSE are given. The new sentence is: *"96% of the NSE using the square root of the*

*daily karstic spring flows are greater than 0.7. Regarding the results of Figure 15c, for rivers in continuous aquifers, 34% of the NSE are greater than 0.7. Moreover 63% of these NSE are greater than 0.5 while 18% are negatives."*

The previous results in the original manuscript were: *"80% of the NSE score using the square root of the daily river flows are greater than 0.8. Regarding the results of Figure 14c, for rivers in continuous aquifers, 27% of the NSE scores are greater than 0.7, 58% are greater than 0.5, and 22% are negatives."*

P25 Figure 1: A new Figure 1 is proposed to present the physical processes included in AquiFR with the associated legend.

P26 Figure 2 : A new Figure 2 is proposed to show the linkage between the different components of AquiFR and the workflow of an AquiFR run.

P39 Figure 15 : Figure 15 was modified in order to account for the 362 gauging stations.

P41 Table 1: A new column is added for specifying if the models were recalibrated or not.

P58 Table 2: Statistical scores were corrected for the Omiécourt piezometer.

**Marked-up manuscript**

This marked-up manuscript does not include the figures but only the legends. The new figures can be found in the revised manuscript.

[revised manuscript text omitted]

---

## Author Response (AR2)

Response to reviewer hess-2019-166
The AquiFR hydrometeorological modelling platform as a tool for improving groundwater resource monitoring over France: evaluation over a 60 year period.

The authors wish to thank the two reviewers for all their corrections, comments and suggestions of improvement for this manuscript. The following text proposes a detailed response for each of them. Bold texts correspond to the author's response.

**Anonymous Referee #1**

- P2L13: includes > include. **Done**
- P6L4: Error in reference. **Done**
- P10L10: newfluxes > new fluxes. **Done**

**Anonymous Referee #2**

1. P1, L32: "Nash-Sutcliffe" **Done**

2. P3, L13: "State scale" **Done**

3. P4, L9-25: I found this part not well placed in the introduction, since it starts describing in details the AquiFR platform, which is the objective of the subsequent section (section 2), with some overlap between these two parts. I suggest moving this part at the beginning of section 2 and removing the redundant information with this section. **The authors agree with this comment. We merged this part with section 2 it in the new revised manuscript.**

4. P4, L13: "and contain" **Done**

5. P5, L32: Remove "Erreur…introuvable" **Done**

6. P6, L4: Remove "Erreur…introuvable" **Done**

7. P6, 26: "assigned to" **Done**

8. P7, L13: "is accounted for" **Done**

9. P7, L26: "dimensional" **Done**

10. P9, L13: "99 km²" **Done**

11. P9, L21-22: Is this use of average pumping estimates realistic? Was this mean cycle calculated over all pumping data available? Would not it be more realistic to take only the first years of the available records and to extend this over the past years? For aquifers where water abstractions have much increased, taking a mean over all the available pumping data may overestimate the abstractions over the most remote parts of the evaluation period. Could the author add a few words on this? **It is not that obvious that the pumping has increased over time. Indeed, pumping from industry was very important in some places in the past, and has decreased significantly. Irrigation varies according to the climate and the equipment. For this reason, it sounds better to impose a mean. The revised manuscript includes a few words about this: "This choice is linked to the lack of knowledge about past pumping. However, we do know that there have been antagonistic**

developments between irrigation and industrial pumping. Irrigation has increased in accordance with the irrigated areas while it varied greatly depending on the climate. Industrial pumping was dominant in the past but has considerably decreased during the past decades (Service de l'observation et des statistiques, 2016)".

Service de l'observation et des statistiques: Repères. L'eau et les milieux aquatiques. Chiffres clés., Ministère de l'Environnement, de l'Energie et de la Mer, 2016.

12. P9, L26: "do not cover" **Done**

13. P10, L3: "new fluxes" **Done**

14. P10, L4: "will be soon updated": any reference to this ongoing work? **It is expected to do it in the next phase of the project, therefore, in the next 2 coming years.**

15. P10, L5: I found it difficult to check all the cited references to get the information on the periods used for calibration. Could the authors add a column in Table 1 to give the period used for calibration in each case? (for each line of the table, if different calibration periods are considered, maybe show the longest period used in each case). This would help the reader assess the level of overlap between the simulation period used in this study and the calibration period. **Thank you for this comment. The revised manuscript includes the calibration period shown directly in the column named Recalibration if necessary.**

16. P10, L10: Similarly to the previous comment, it would be useful for the reader if the authors could add an extra column in Table 1 giving the type of criteria used for calibration (and possibly the various variables – piezometric level and/or flows). This would show the various strategies used in these past works. **An extra column is added to specify the type of variables used for recalibrated the models. The following text is added to the legend: "Periods of calibration are given in the Recalibration column and the type of variables used for recalibration in the Variables column. GW means groundwater level and RF river flow. GW levels were evaluated using RMSE and bias criteria. River flows were evaluated using NSE and ratio criteria."**

17. P10, L29: "relative": the formulation is not expressed in relative but absolute terms **Done**

18. P10, L31: In the equations, the summation symbol refers to i index but then the variable X is referring to t. Maybe use instead Q(i) to be more consistent in notations. Change notations accordingly for the other equations. Maybe remind that BIAS has the same unit as X and that the perfect value is 0, with negative values corresponding to overestimation and positive values corresponding to underestimation by the model (or maybe there is an error in Eq.1, see also comment # 25 on this issue). **Done**

19. P11, top: Please give a few words on how the values of the NRMSE_BE criterion should be interpreted (what could be considered a good NMRSE_BE value?). **A comment is added to the text: "The NRMSE_BE criterion is always positive and starts from 0 for a perfect simulation of the observed amplitudes. A NRMSE_BE lower than 0.8 can be considered as a reasonable estimation of the temporal evolution of the observed water table."**

20. P11, L11: I found this sentence unclear. What do you mean? **The authors agree. NSE is strongly affected by biases between observation and simulation. Therefore, a bias value that can be considered as reasonable (i.e. < 1 m) can lead to poor NSE scores. If the reviewer agree, we replace this sentence in the revised manuscript by "Its use for comparing groundwater levels is less obvious regarding its strong sensitivity to the biases between observation and simulation.**

21. P12, L8-21: This part is not well placed in the Results section, since it presents the data used. I suggest moving it to a new subsection on data at the end of section 3. **The authors agree. This part is moved in a new section « 3.4 Dataset and model setup » in the revised manuscript.**

22. P12, L8-21: I did not find information on the way model warm-up is made at the beginning of the simulation period. I guess this can be an issue and a potential source of error. Could the authors give some information on this? **Initial conditions correspond to a proxy year of the 1958 year that was defined using time series of long-term observed groundwater levels starting from 1958. A sentence about this is included in the new subsection 3.4 Data : "State variables from August 1, 2013 were chosen from a first simulation over the 1958-2018 period in order to initialize the simulation in August 1, 1958. The 2013 year was chosen as the best proxy year of the 1958 year by analysing long-term time series of observed groundwater levels having data since 1958."**

23. P12, L12: "a few measurements" **Done**

24. P12, L20: "not yet fully available" **Done**

25. P12, L23-24: From equation (1), a positive value would mean an underestimation. **Thank you. This sentence was right but there was a mistake in the description of Equation 1. It is now corrected in the revised manuscript.**

26. P12, L26: instead of "accumulated distribution", use "cumulative distribution". Change accordingly elsewhere in the text and figures (graphs and captions in Fig. 6, 7, 15) **Done**

27. P12, L29-30: How these values can be interpreted? (see comment #19) **Section 3.3 includes now a few word about this (see answer to comment #19).**

28. P13, L6-7: The observations show annual cycles starting from 2008. Is there any change in the measurement device? **Yes, observed values were monitored monthly until 1995, then weekly until 2008, and then daily after this date.**

29. P13, L8 and 11: "Helloin" is written with uppercase elsewhere in the article. **Done**

30. P13, L12: This period seems also badly simulated for the Omiécourt and Farceaux case studies, which are all in the same region. Is it just a coincidence or were there specific climatic conditions on this period, which are not well captured by the models? **Thank you for this remark. The authors agree about this comment. The fact that these biases occur in different regions with different hydrogeological models suggests indeed an underestimation of recharge from SURFEX. However, errors in the hydrogeological models cannot be excluded. A specific analysis of these fluxes that is beyond the scope of this paper could be useful to explain these biases.**

31. P13, L24-26: I am not sure these two figures are really useful. Very little is said on these figures and they do not seem to convey different information. I would remove the figure on correlation. **The authors agree to the fact that the Figure on Correlation brings little information. We keep only the NSE Figure in the revised manuscript, as suggested by the reviewer.**

32. P14, L29: "Kling-Gupta efficiency" **Done**

33. P15, L13: "negative" **Done**

34. P15, L19: Not sure this comment is fully consistent with what is said on top of page 13. Please clarify this. **Thank you for this remark. This was an error. We meant poor bias scores instead of poor NRMSE-BE scores for the Loire River. This comment is now consistent with top of page 13.**

35. P16, L18: "in section 3.2" **Done**

36. P16, L17-19: Generally, testing a model outside calibration range is considered a good way to evaluate its extrapolation capacity and robustness. Here it is difficult to evaluate this since the calibration and evaluation periods overlap. The authors could discuss this point. Is there some perspective to more thoroughly evaluate the robustness of model proposed in AquiFR? **According to the period of calibration of each model now described in the new Table 1 of the revised manuscript, no calibration was undertaken before 1986 and after 2015 whatever the model used. Thus, in a few years, we will have several independent years after 2015 with observations to consider a better evaluation of the robustness of the model.**

37. Fig. 6: Could the cumulative distribution give also the value for an absolute bias lower than 1, to make a better correspondence with the scale of the map above (in fig. a). **Done**

38. Fig. 12, caption: "for June 2016 for" **Done**

39. Fig. 13: It seems that the observations are not available on the whole period in each case. But it is a bit difficult to distinguish between the case where observed and simulated values perfectly overlap from the case where there are gaps in the observations. I suggest adding the information on periods of gaps in the observations, e.g. by putting a grey background on the graphs for the corresponding periods. **Done**

40. Fig. 13: in the caption: "dashed blue" **Done**

41. Fig. 14: Same comments as #39 and 40 **Done**

42. Table 4: The information contained in the table could be directly introduced in Fig. 14 (in brackets after the name of the catchment, as done in Fig. 13). This would reduce the number of tables. **Done**

**List of all relevant changes**

All the relevant changes made to the manuscript correspond to the technical corrections previously stated by the reviewers as well as complements that were arise by the reviewers, including:

- A paragraph of the Introduction that was moved and merged to Section 2: The AquiFR Hydrometeorological Modelling Platform
- The introduction of the Section 4 Results that is moved to a new subsection Section 3.4 Dataset and model setup
- Figure 10 and Table 4 were deleted according to the suggestions of reviewer #2.

**Marked-up manuscript**

[revised manuscript text omitted]

---

## Author Response (AR3)

Response to reviewer hess-2019-166

The AquiFR hydrometeorological modelling platform as a tool for improving groundwater resource monitoring over France: evaluation over a 60 year period.

The authors wish to thank the editor for its careful review. The list of all relevant changes include the response to the comments regarding the technical corrections required for the manuscript.

**List of all relevant changes**

All the relevant changes made to the manuscript correspond to technical corrections in order to avoid multi-letter variable names. It includes:

- BIAS variable name changed in $B$ (Equation (1))
- NRMSE_BE variable name changed in $E_{\mathrm{NRMS\_BE}}$ (Equation (2))
- NSE variable name changed in $E_{\mathrm{f}}$ (Equation (3))
- Ratio variable name changed in $R_{\mathrm{d}}$ (Equation (4))

These new variable names replace the old variable names in the entire manuscript, including Figure 7, 9, 12, 13 and 14. Moreover, subscripts in variables names are now in normal font (such as $X_{\mathrm{sim}}(t)$), in particular in Section 3.3.

**Marked-up manuscript**

[revised manuscript text omitted]

$$E_{\mathrm{NRMS\_BE}}NRMSE\_BE = \frac{1}{\sigma_{\mathrm{obs}}}\sqrt{\frac{\sum_{t=1}^{n}[(X_{\mathrm{sim}}(t)-\overline{X_{\mathrm{sim}}})-(X_{\mathrm{obs}}(t)-\overline{X_{\mathrm{obs}}})]^2}{n}}$$

5          (2)

with $\overline{X_{\mathrm{sim}}}$ the temporal mean of simulated values over the considered period and $\sigma_{\mathrm{obs}}$ the observed standard deviation. The $E_{\mathrm{NRMS\_BE}}$ criterion is always positive and starts from 0 for a perfect simulation of the observed amplitudes. A $E_{\mathrm{NRMS\_BE}}$ criterion lower than 0.8 can be considered as a reasonable estimation of the temporal evolution of the observed water table.

10   The Nash-Sutcliffe model Efficiency coefficient $E_t$ (Nash and Sutcliffe, 1970) measures the variance between the observed and simulated values. It is often applied to compare observed and simulated river flows but can be used for other variables. Its use for comparing groundwater levels is less obvious regarding its strong sensitivity to the biases between observation and simulation. It is equal to 1 when the model fits perfectly the observations. A $E_t$ criterion above 0.7 is generally accepted as a good estimate of the signal dynamic, however depending on the hydrogeological and climate context

15   of the basin. A negative $E_t$ value means that the mean observed signal is a better predictor than the model.  $E_t$ is calculated as follows:

$$\mathrm{NSE}_tE = 1 - \frac{\sum_{t=1}^{n}(X_{\mathrm{obs}}(t)-X_{\mathrm{sim}}(t))^2}{\sum_{t=1}^{n}(X_{\mathrm{obs}}(t)-\overline{X_{\mathrm{obs}}})^2},$$
(3)

with $\overline{X_{\mathrm{obs}}}$ the temporal mean of observed values over the considered period.

The annual discharge ratio $R_d$ criterion helps to compare the mean simulated and observed river flows as follows:

20   $R_d Ratio = \frac{\overline{Q_{\mathrm{sim}}}}{\overline{Q_{\mathrm{obs}}}},$
(4)

[revised manuscript text omitted]